# Geometric Exploration for Online Control

**Orestis Plevrakis**
Princeton University
orestisp@princeton.edu

**Elad Hazan**
Princeton University
Google AI Princeton
ehazan@princeton.edu

## Abstract

We study the control of an *unknown* linear dynamical system under general convex costs. The objective is minimizing regret vs the class of strongly-stable linear policies. In this work, we first consider the case of known cost functions, for which we design the first polynomial-time algorithm with $n^3\sqrt{T}$-regret, where $n$ is the dimension of the state plus the dimension of control input. The $\sqrt{T}$-horizon dependence is optimal, and improves upon the previous best known bound of $T^{2/3}$. The main component of our algorithm is a novel geometric exploration strategy: we adaptively construct a sequence of barycentric spanners in an over-parameterized policy space. Second, we consider the case of bandit feedback, for which we give the first polynomial-time algorithm with $poly(n)\sqrt{T}$-regret, building on Stochastic Bandit Convex Optimization.

## 1 Introduction

In this paper we study the online control of an unknown linear dynamical system under general convex costs. This is a fundamental problem in control theory and it also embodies a central challenge of reinforcement learning: balancing exploration and exploitation in continuous spaces. For this reason, it has recently received considerable attention from the machine learning community.

**Controlling unknown LDS:** In a linear dynamical system (LDS), the system state $x_t \in \mathbb{R}^{d_x}$ evolves as

$$x_{t+1} = A_* x_t + B_* u_t + w_t, \quad \text{where } x_1 = 0, \tag{1}$$

$u_t \in \mathbb{R}^{d_u}$ is the learner's control input, $w_t \in \mathbb{R}^{d_x}$ is a noise process drawn as $w_t \overset{\text{i.i.d}}{\sim} N(0, I)$, and $A_*, B_*$ are unknown system matrices. The learner applies control $u_t$ at timestep $t$, then observes the state $x_{t+1}$ and suffers cost $c(x_t, u_t)$, where $c$ is a convex function. We consider two forms of cost information for the learner: the case where $c$ is known in advance, and the bandit version where only the scalar cost is observed.

Even if the dynamics where known, there are problem instances where the optimal policy is a very complicated function [6]. A way to circumvent this is to consider a policy class that is both expressive and tractable, and aim for performing as well as the best policy from that class. The objective that captures this goal is regret, and is a standard performance metric in online control. In this paper, in accordance with the previous works, we choose the policy class to be all *strongly-stable linear policies*[1]. These are policies that 1) select the control as a linear function of the state, and 2) mix fast to a steady state. These policies are a reasonable baseline, since they are optimal for the special case where the cost $c$ is strongly convex quadratic.

Formally, regret with respect to the class $\mathcal{K}$ of all strongly stable linear policies is defined as

$$R_T = \sum_{t=1}^{T} c(x_t, u_t) - T \min_{K \in \mathcal{K}} J(K), \tag{2}$$

where a policy $K \in \mathcal{K}$ applies $u_t = Kx_t$, and, letting $\mathbb{E}_K$ denote expectation under this policy,

$$J(K) = \lim_{T \to \infty} \frac{1}{T} \mathbb{E}_K \left[ \sum_{t=1}^{T} c(x_t, u_t) \right] \tag{3}$$

is the average infinite-horizon cost of $K$.

Our main result is a polynomial-time algorithm for the case of known cost function, that achieves $n^3\sqrt{T}$-regret, where $n = d_x + d_u$. This is the optimal dependence in the time-horizon and our result improves upon the previous best known bound of $T^{2/3}$ [17, 29]. Perhaps more importantly than the regret bound is that, using ideas from convex geometry, we design a novel exploration strategy which significantly expands the existing algorithmic toolbox for balancing exploration and exploitation in linear dynamical systems, as we explain below.

**Beyond explore-then-commit: the challenge of exploration**

The only algorithms we know for this problem apply the simplest exploration strategy: "explore-then-commit" (ETC), known in control literature as certainty equivalence. In ETC, the learner spends the first $T_0$ steps playing random controls (e.g. $u_t \sim N(0, I)$), then estimates the system dynamics, and thereafter executes a greedy policy, based on these estimates. On the other hand, the whole stochastic bandit and RL theory literature is about sophisticated and sample-efficient exploration, mostly relying on the principle of *optimism in the face of uncertainty* (OFU). Unfortunately, implementing OFU in online control requires solving optimization problems that are intractable in general [1]. Even though this computational issue can be circumvented for the case of quadratic costs using semidefinite programming [12], these techniques do not apply for general convex costs. In this work, we do not follow the OFU principle. Our exploration strategy is based on adaptively constructing a sequence of barycentric spanners (Definition 7) in an over-parameterized policy space.

**The importance of general convex costs**

The special case of convex quadratic costs is the classical linear quadratic regulator and is frequently used, because it leads to a nice analytical solution when the system is known [7]. However, this modeling choice is fairly restrictive, and in 1987, Tyrrell Rockafellar [23] proposed the use of general convex functions for modelling the cost in a LDS, in order to handle constraints on state and control. In practice, imposing constraints is crucial for ensuring safe operating conditions.

## 1.1 Statement of results.

We consider both the setting where $A_*$ is strongly stable (Assumption 1), and in the Appendix, we deal with unstable systems, by assuming that the the learner is initially given a strongly-stable linear policy (Assumption 4)[2]. Our main result is the geometric exploration strategy given in Algorithm 1, for the case of known cost function. Algorithm 4 is for the case of bandit feedback. We now state informal versions of our theorems. Let $C = C(A_*, B_*)$ denote a constant that depends polynomially on natural system parameters.

**Theorem 1** (informal)**.** *For online control of LDS with known cost function, with high probability, Algorithm 1 has regret* [3]

$$R_T \leq \widetilde{O}(C) \cdot n^3 \sqrt{T}. \tag{4}$$

**Theorem 2** (informal)**.** *For online control of LDS with bandit feedback, with high probability, Algorithm 4 has regret*

$$R_T \leq \widetilde{O}(C) \cdot poly(n)\sqrt{T}. \tag{5}$$

In Theorem 2, the polynomial dependence in $n$ is rather large ($n^{36}$). The large dimension dependence is typical in $\sqrt{T}$-regret algorithms for bandit convex optimization (BCO). Our setting is even more challenging than BCO, since the environment has a state.

## 1.2 Our Approach and New Techniques

The cost $J(K)$ is nonconvex in $K$. To remedy this, we use an alternative disturbance-based policy class, introduced in [4], where the control is linear in the past disturbances (as opposed to the state):

$$u_t = \sum_{i=1}^{H} M^{[i-1]} w_{t-i}, \tag{6}$$

where $H$ is a hyperparameter. This parameterization is useful, because the offline search for an optimal disturbance-based policy can be posed as a convex program. In [4], the authors show that disturbance-based policies can approximate all strongly stable linear policies. We move on to describe the novel components of our work.

**Geometric exploration:** Algorithm 1 runs in epochs and follows the phased-elimination paradigm [20]. During epoch $r$, it focuses on a convex disturbance-based policy set $\mathcal{M}_r$, which by the end of the epoch will be substituted by $\mathcal{M}_{r+1} \subseteq \mathcal{M}_r$. In the beginning of the epoch, it decides on a set of *exploratory policies* to be executed during the epoch. This is done by constructing a barycentric spanner of the policy set $\mathcal{M}_r$ (Definition 7). At the end of the epoch, it refines its estimates for the system matrices $A_*, B_*$ and creates $\mathcal{M}_{r+1} \subseteq \mathcal{M}_r$.

**Estimating the disturbances:** A typical difficulty in using disturbance-based policies is that we do not know the disturbances, and since the system matrices are unknown, we cannot recover $w_t$ by using $w_t = x_{t+1} - A_* x_t - B_* u_t$. A key contribution of this work is showing that *disturbances can be accurately estimated, without accurately estimating the system matrices* $A_*, B_*$.

**Coupling:** At the end of epoch $r$, we create estimates $\widehat{A}, \widehat{B}$ of $A_*, B_*$, which we use to create the next policy set $\mathcal{M}_{r+1}$. These estimates are not necessarily accurate in every direction. To argue that $\mathcal{M}_{r+1}$ contains good policies, we employ a probabilistic coupling argument between the true system and the estimated one. This could be a useful technique for future works.

**Robustness of stochastic BCO:** For bandit feedback, we require a generalization of the Stochastc Bandit Convex Optimization (SBCO) framework, where the noise contains a small adversarial component in addition to the stochastic part. The need for robustness to adversarial noise arises from the estimation error on the disturbances $\|\widehat{w}_t - w_t\|$. We prove that by appropriately changing the parameters in the SBCO algorithm from [3], we can get $\sqrt{T}$-regret for this more general model.

## 1.3 Prior Work

**LQR:** When the cost $c$ is convex quadratic, we obtain the online linear quadratic regulator (LQR) [1, 14, 21, 12, 27]. The problem was introduced in [1], and [21, 12, 27] gave $\sqrt{T}$-regret algorithms with polynomial runtime and polynomial regret dependence on relevant problem parameters. In [27], the authors proved that $\sqrt{T}$-regret is optimal.

**Convex costs:** Closer to our work are recent papers on online control with general convex costs [17, 29, 19]. These papers consider even more general models, i.e, adversarially changing convex cost functions, in [17, 29] they also allow adversarial disturbances and [29, 19] deal with partial observation. Furthermore, all the considered policy classes have linear structure and in [17] they use the same policy class as we do, i.e. strongly-stable linear policies. Despite the differences in the models, a common feature of these works is that all algorithms apply explore-then-commit (ETC). For our setting, ETC gives $T^{2/3}$-regret. Under the assumption that $c$ is strongly convex, the problem is significantly simplified and ETC achieves $\sqrt{T}$-regret [29].

**Linear system identification:** To address unknown systems, we make use of least-squares estimation [28, 24]. Recent papers deal with system indentification under partial observation [22, 25, 30, 26].

**Bandit feedback:** Control with bandit feedback has been studied in [9] (known system) and [15] (both known and unknown system). Our result is comparable to [15], and improves upon the $T^{3/4}$ regret bound that they achieve, when the disturbances are stochastic and the cost is a fixed function.

**Barycentric spanners:** Barycentric spanners have been used for exploration in stochastic linear bandits [5]. However, in that context, the barycentric spanner is computed offline and remains fixed, while our algorithm adaptively changes it, based on the observed states.

## 2 Setting and Background

**Notation.** For a matrix $A$, we use $\|A\|$ to denote its spectral norm, and for a positive semidefinite matrix $\Sigma \succcurlyeq 0$, we use $\|A\|_\Sigma$ to denote $\sqrt{\mathrm{tr}(A^T \Sigma A)}$. For notational simplicity, "with high probability" means with probability at least $1 - 1/T^c$, where $c$ is a sufficiently large constant.

### 2.1 Assumptions and Regret Benchmark

Linear policies are parameterized with a matrix K and apply controls $u_t = Kx_t$. We define the class of strongly stable linear policies. This definition was introduced in [11] and quantifies the classical notion of a stable policy in manner that permits non-asymptotic regret bounds.

**Definition 3.** A linear policy $K$ is $(\kappa, \gamma)$-strongly stable if there exists a decomposition of $A_* + B_* K = Q\Lambda Q^{-1}$ with $\|\Lambda\| \leq 1 - \gamma$ and $\|K\|, \|Q\|, \|Q^{-1}\| \leq \kappa$.

For simplicity, in the main text we assume that $K = 0$ is strongly stable.

**Assumption 1.** *The linear policy $K = 0$ is $(\kappa, \gamma)$-strongly stable, for some known constants $\kappa \geq 1$, $\gamma \geq 0$.*

When referring to this assumption, we will also say "$A_*$ is $(\kappa, \gamma)$-strongly stable". In Appendix A, we relax this assumption and consider possibly unstable $A_*$, by using an initial stabilizing policy (Assumption 4). Assumption 4 is standard in online control literature (e.g., [12, 17, 8]), and without it, the regret is exponential in $n$ ([10]). The next assumptions are that $B_*$ is bounded and the cost $c$ is Lipschitz.

**Assumption 2.** *The norm $\|B_*\| \leq \beta$, for some known constant $\beta \geq 1$.*

**Assumption 3.** *The cost function $c$ is $1$-Lipschitz.* [4]

Let $\mathcal{K}$ be the set of all $(\kappa, \gamma)$-strongly stable linear policies. Our objective is regret with respect to the policy class $\mathcal{K}$, as given in equation 2.

#### 2.1.1 Disturbance-based policies

We provide some background definitions, which were introduced in [4].

**Definition 4.** A disturbance-based policy $M = \left(M^{[0]}, \ldots, M^{[H-1]}\right)$ applied at time $t$ chooses control $u_t = \sum_{i=1}^{H} M^{[i-1]} w_{t-i}$, where $w_s = 0$, for $s \leq 0$.

The advantage of disturbance-based policies is that the average infinite horizon cost $J(M)$ of the policy $M = \left(M^{[0]}, \ldots, M^{[H-1]}\right)$ is a convex function of $M$. Also, this comes for free in terms of expressiveness, since disturbance-based policies can approximate all policies in $\mathcal{K}$ (Theorem 5 below). Formally, let $H := C \cdot \gamma^{-1} \cdot \log\left(T d_x d_u \kappa \beta \gamma^{-1}\right)$, where $C$ is a sufficiently large constant (e.g., 1000), and $\mathcal{M} = \left\{ \left(M^{[0]}, \ldots, M^{[H-1]}\right) \,\middle|\, \|M^{[i]}\| \leq \kappa^3 \beta (1-\gamma)^i \right\}$.

**Theorem 5** ([4]). *For all $K \in \mathcal{K}$, there exists a disturbance-based policy $M \in \mathcal{M}$, such that $\left| J(M) - J(K) \right| \leq 1/T$.*

This theorem is a slight modification of a result in [4], and we give its proof in Appendix G.3. Under the execution of a disturbance-based policy $M$, the state can be expressed as $x_{t+1} = A_*^{H+1} x_{t-H} +$

$\sum_{i=0}^{2H} \Psi_i(M \mid A_*, B_*) w_{t-i}$ [5], where $\Psi_i(M \mid A_*, B_*)$ are affine functions of $M$. We provide exact expressions for $\Psi_i$ in Appendix H. Because $A_*$ is $(\kappa, \gamma)$-strongly stable, it can be shown that $A_*^{H+1} \approx 0$. This leads to the definition of two time-independent quantities: surrogate state and surrogate control.

**Surrogate state and control:** Let $\eta = (\eta_0, \eta_1, \dots \eta_{2H})$, where $\eta_i \overset{i.i.d}{\sim} N(0, I)$. We define $u(M \mid \eta) = \sum_{i=0}^{H-1} M^{[i]} \eta_i$ and $x(M \mid A_*, B_*, \eta) = \sum_{i=0}^{2H} \Psi_i(M \mid A_*, B_*) \eta_i$. As we mentioned, $A_*^{H+1} \approx 0$, so when we execute policy $M$ and $t \geq \Omega(H)$, the state/control pair $(x_t, u_t)$ is almost identically distributed with $\left( x(M \mid A_*, B_*, \eta), u(M \mid \eta) \right)$.

**Convex surrogate cost:** We now define a function that approximates the cost $J(M)$, without involving infinite limits that create computational issues. Let

$$\mathcal{C}(M \mid A_*, B_*) := \mathbb{E}_\eta \left[ c\Big( x(M \mid A_*, B_*, \eta), \, u(M \mid \eta) \Big) \right]. \tag{7}$$

The cost $\mathcal{C}$ is convex, because $x$ and $u$ are affine in $M$ and $c$ is convex. The following theorem establishes that $J(M)$ is almost equal to $\mathcal{C}(M \mid A_*, B_*)$.

**Theorem 6** ([4]). *For all $M \in \mathcal{M}$, we have $\left| \mathcal{C}(M \mid A_*, B_*) - J(M) \right| \leq 1/T$.*

Again, this theorem is almost proved in [4], and for completeness, we provide its proof in Appendix G.4. The algorithms we present in the paper aim to minimize $\mathcal{C}(M \mid A_*, B_*)$. The difficulty is that we do not know this function, since we do not know $A_*, B_*$.

### 2.1.2 Barycentric spanners

Barycentric spanners and approximate barycentric spanners were introduced in [5]. Here, we will use the affine[6] version of approximate barycentric spanners, defined below.

**Definition 7.** Let $K$ be a compact set in $\mathbb{R}^d$. A set $X = \{x_0, x_1, \dots, x_d\} \subseteq K$ is a $C$-approximate affine barycentric spanner for $K$ if every $x \in K$ can be expressed as $x = x_0 + \sum_{i=1}^d \lambda_i (x_i - x_0)$, where the coefficients $\lambda_i \in [-C, C]$.

The above theorem follows from Proposition 2.5 in [5], as we show in Appendix G.5.

**Theorem 8.** *Suppose $K \subseteq \mathbb{R}^d$ is compact and not contained in any proper affine subspace. Given an oracle for optimizing linear functions over $K$, for any $C > 1$ we can compute a $C$-approximate affine barycentric spanner for $K$ in polynomial time, using $O(d^2 \log_C(d))$ calls to the oracle.*

## 3 Known cost function

In this section, we present our polynomial-time algorithm for the case of known cost function (Algorithm 1), we state our main theorem and give an overview of its proof. Formally, known cost function means that the algorithm has offline access to the value $c(x, u)$ and gradient $\nabla_{x,u} c(x, u)$ for all state/control pairs $(x, u)$.

### 3.1 Algorithm and statement of main theorem

Algorithm 1 receives as input some initial estimates $A_0, B_0$ that approximate the true system matrices $A_*, B_*$ within error $\epsilon$. As we state in Theorem 9, $\epsilon$ needs to be $1/\mathrm{poly}(d_x, d_u)$, and we can make sure this is satisfied by executing a standard warm-up exploration procedure, given in Appendix F. Moreover, our algorithm computes a new 2-approximate affine barycentric spanner in the beginning of each epoch. This step can be implemented in polynomial time, as we explain in Subsection 3.3.

**Algorithm 1:** Geometric Exploration for Control

---

**Input:** estimates $A_0, B_0$ satisfying $\|(A_0\ B_0) - (A_*\ B_*)\|_F \leq \epsilon$.

Initialize policy set $\mathcal{M}_1 = \mathcal{M}$, matrix estimates $(\widehat{A}_1\ \widehat{B}_1) = (A_0\ B_0)$, disturbance estimates
$\ \widehat{w}_t = 0$ for $t \leq 0$.

Set $d = d_u \cdot d_x \cdot H$.

Set $t = 1$ and observe $x_1$.

**for** $r = 1, 2, \ldots$ **do**
   Set $\epsilon_r = 2^{-r}$.
   Compute a 2-approximate affine barycentric spanner of $\mathcal{M}_r$: $\{M_{r,0}, M_{r,1}, \ldots, M_{r,d}\}$.
   Set $T_r = \widetilde{\Theta}(\kappa^4 \gamma^{-3}) \cdot \epsilon_r^{-2} \cdot d_x d_u (d_x + d_u)^2$. [a]
   Call Execute-Policy($M_{r,0},\ d \cdot T_r$).
   **for** $j = 1, \ldots, d$ **do**
      Call Execute-Policy($M_{r,j},\ T_r$).
   **end**
   Set $t_r = t$ (current timestep).
   Eliminate suboptimal policies: [b]

$$\mathcal{M}_{r+1} = \left\{ M \in \mathcal{M}_r \ \middle|\ \mathcal{C}\left(M\ \middle|\ \widehat{A}_{t_r}, \widehat{B}_{t_r}\right) - \min_{M' \in \mathcal{M}_r} \mathcal{C}\left(M'\ \middle|\ \widehat{A}_{t_r}, \widehat{B}_{t_r}\right) \leq 3\epsilon_r \right\} \quad (8)$$

**end**

---

[a] The $\widetilde{\Theta}(\cdot)$ in the definition of $T_r$ denotes $T_r = C \cdot \kappa^4 \gamma^{-3} \cdot \epsilon_r^{-2} \cdot d_x d_u (d_x + d_u)^2$, for some sufficiently large $C = polylog(T)$. The same applies for the definition of $\lambda$ in Algorithm 3.

[b] Equation 8 is a definition, not a step that requires computation. The computational step related to $\mathcal{M}_{r+1}$ is the construction of the 2-approximate affine barycentric spanner in the beginning of the next epoch.

---

**Algorithm 2:** Execute-Policy

---

**Input:** Policy $M$ and execution-length $L$.

**for** $s = 1, 2, \ldots, L$ **do**
   Apply control $u_t = \sum_{i=1}^{H} M^{[i-1]} \widehat{w}_{t-i}$.
   Observe $x_{t+1}$.
   Call System-Estimation, to get $\widehat{A}_{t+1}, \widehat{B}_{t+1}$.
   Record the estimate $\widehat{w}_t = x_{t+1} - \widehat{A}_{t+1} x_t - \widehat{B}_{t+1} u_t$.
   Set $t = t + 1$.
**end**

---

**Algorithm 3:** System-Estimation

---

Set $\lambda = \widetilde{\Theta}\left(\kappa^{10} \beta^4 \gamma^{-5}\right) \cdot d_x d_u (d_x + d_u)^3$.

Let $(\widehat{A}_{t+1}\ \widehat{B}_{t+1})$ be a minimizer of

$$\sum_{s=1}^{t} \|(A\ B)z_s - x_{s+1}\|^2 + \lambda \|(A\ B) - (A_0\ B_0)\|_F^2$$

over all $(A\ B)$, where $z_s = \begin{pmatrix} x_s \\ u_s \end{pmatrix}$.

---

We now formally state our main theorem.

**Theorem 9.** *Let* $C_1 = \kappa^{10} \beta^4 \gamma^{-5}$, $C_2 = \kappa^{14} \beta^4 \gamma^{-8}$ *and* $C_3 = \kappa^5 \beta^2 \gamma^{-2.5}$. *Suppose that the initial estimation error bound* $\|(A_0\ B_0) - (A_*\ B_*)\|_F \leq \epsilon$ *satisfies* $\epsilon^2 \leq (C_1 \cdot d_x d_u (d_x + d_u))^{-1}$. *Assume* $T \geq \widetilde{\Omega}(C_2) \cdot d_x d_u (d_x + d_u)^5$. *Then, with high probability, Algorithm 1 satisfies*

$$R_T \leq \widetilde{O}(C_3) \cdot d_x d_u (d_x + d_u) \sqrt{T}. \quad (9)$$

In [12], the authors analyze the warm-up exploration (Algorithm 6 in Appendix F). In Appendix F, we show that their analysis can be combined with Theorem 9 to show the following.

**Corollary 10.** *Let $C_2 = \kappa^{14}\beta^4\gamma^{-8}$ and $C_4 = \kappa^{13}\beta^5\gamma^{-5.5}$. Assume $T \geq \widetilde{\Omega}(C_2) \cdot d_x d_u (d_x + d_u)^5$. If we run the warm-up exploration and then run Algorithm 1, then with high probability,*

$$R_T \leq \widetilde{O}(C_4) \cdot d_x d_u (d_x + d_u)\sqrt{T}. \tag{10}$$

## 3.2   Proof overview

This section gives an overview of the proof of Theorem 9, while formal statements and proofs in the Appendix. The main component of our proof is bounding the *stationary regret*, i.e.,

$$R_T^{st} = \sum_{t=1}^{T} \mathcal{C}\left(M_t \mid A_*, B_*\right) - T \cdot \mathcal{C}\left(M_* \mid A_*, B_*\right), \tag{11}$$

where $M_t$ is the policy executed at time $t$ and $M_* \in \arg\min_{M \in \mathcal{M}} \mathcal{C}(M \mid A_*, B_*)$. Theorem 9 follows from the following two lemmas.

**Lemma 11.** *With high probability, $R_T - R_T^{st} \leq \widetilde{O}\left(\kappa^5\beta^2\gamma^{-2.5}\right) \cdot (d_x + d_u)^{3/2}\sqrt{T}$.*

**Lemma 12.** *With high probability, $R_T^{st} \leq \widetilde{O}\left(\kappa^2\gamma^{-2}\right) \cdot d_x d_u (d_x + d_u)\sqrt{T}$.*

The proof of Lemma 11 is mostly technical and we do not discuss it in the main text. We will focus instead on the proof of Lemma 12. First, observe that the policies $M_t$ executed during epoch $r$ belong to $\mathcal{M}_r$, since they are elements of a 2-approximate affine barycentric spanner of $\mathcal{M}_r$. To bound $R_T^{st}$, we show that with high probability, for all policies $M \in \mathcal{M}_r$, the suboptimality gap $R^{st}(M) := \mathcal{C}\left(M \mid A_*, B_*\right) - \mathcal{C}\left(M_* \mid A_*, B_*\right) \leq O(2^{-r})$, from which Lemma 12 follows after some calculations. To bound this suboptimality gap, we show that the elimination step of Algorithm 1 (Equation 8) is effective, i.e., it removes only the $\Omega(2^{-r})$-suboptimal policies ($R^{st}(M) \geq \Omega(2^{-r})$). This effectiveness comes from the following bound on the estimation error $\left|\mathcal{C}(M \mid \widehat{A}_{t_r}, \widehat{B}_{t_r}) - \mathcal{C}(M \mid A_*, B_*)\right|$.

**Lemma 13.** *With high probability, for all epochs $r$ and for all $M \in \mathcal{M}_r$, we have*

$$\left|\mathcal{C}(M \mid \widehat{A}_{t_r}, \widehat{B}_{t_r}) - \mathcal{C}(M \mid A_*, B_*)\right| \leq 2^{-r}. \tag{12}$$

The proof of this lemma is the crux of our analysis, employs all our new techniques and is done in two steps. The first step bounds the estimation error on the cost (LHS of 12) by the estimation error on the matrices, with respect to a matrix norm associated with policy $M$. More specifically, let $\Sigma(M)$ be the covariance matrix of the random vector $z\left(M \mid A_*, B_*, \eta\right)$ defined as

$$z\left(M \mid A_*, B_*, \eta\right) := \begin{pmatrix} x\left(M \mid A_*, B_*, \eta\right) \\ u\left(M \mid \eta\right) \end{pmatrix}, \tag{13}$$

where $x\left(M \mid A_*, B_*, \eta\right)$, $u(M \mid \eta)$ and the distribution of $\eta$ are given in Subsection 2.1.1. In other words, $\Sigma(M) = \mathbb{E}_\eta\left[z\left(M \mid A_*, B_*, \eta\right) \cdot z\left(M \mid A_*, B_*, \eta\right)^T\right]$. We prove the following lemma.

**Lemma 14** (informal). *Let $\Delta_{t_r} = (\widehat{A}_{t_r}\ \widehat{B}_{t_r}) - (A_*\ B_*)$. For all $M \in \mathcal{M}_r$,*

$$\left|\mathcal{C}(M \mid \widehat{A}_{t_r}, \widehat{B}_{t_r}) - \mathcal{C}(M \mid A_*, B_*)\right| \leq O\left(\kappa^2\gamma^{-1}\right) \cdot \left\|\Delta_{t_r}^T\right\|_{\Sigma(M)}. \tag{14}$$

The proof idea here is to analyze two coupled dynamical systems: the first system has system matrices $\widehat{A}_{t_r}, \widehat{B}_{t_r}$ and the second has $A_*, B_*$. The coupling is done by applying the same controls and disturbances to both of them. The formal statement and proof are in Appendix B.1 and B.2. The second step is the following lemma. Let $C$ be a large constant.

**Lemma 15.** *(informal) With high probability, $\|\Delta_{t_r}^T\|_{\Sigma(M)} \leq (C^{-1}\kappa^{-2}\gamma) \cdot 2^{-r}$, for all $r$, $M \in \mathcal{M}_r$.*

This lemma completes the proof of Lemma 13, by choosing a sufficiently large $C$. To prove Lemma 15, we first focus on the exploratory policies $\mathcal{M}_{r,j}$ and for notational convenience, we define $M_{r,d+1} = M_{r,d+2} = \cdots = M_{r,2d} = M_{r,0}$. We show that with high probability, $\sum_{j=1}^{2d} \|\Delta_{t_r}^T\|_{\Sigma(M_{r,j})}^2 \leq (C_1^{-1}\gamma^2\kappa^{-4}d^{-1}) \cdot 2^{-2r}$, where $d = d_x d_u H$ and $C_1$ is a large constant. To finish the proof, we show that for all $M \in \mathcal{M}_r$, we can express $M$ as an affine combination of $\{M_{r,j}\}_{j=1}^{2d}$ using coefficients in $[-3, 3]$. Then, we show that this implies $\Sigma(M) \preccurlyeq 18d \cdot \sum_{j=1}^{2d} \Sigma(M_{r,j})$, which completes the proof. The formal statement and proof are in Appendix B.1 and B.3.

**Estimating the disturbances:** A challenge in proving both Lemmas 12 and 11 comes from using the estimates of the disturbances $(\widehat{w}_t)_t$, instead of the actual ones $(w_t)_t$. We deal with this via proving that $\widehat{w}_t$ are on average very accurate.

**Lemma 16.** *With high probability, we have* $\sum_{t=1}^{T} \|\widehat{w}_t - w_t\|^2 \leq \widetilde{O}(1) \cdot (d_x + d_u)^3$. [7]

We provide the proof in Appendix B.4. A notable aspect of that proof is that it does not use any information about the way we choose the controls $u_t$. On the other hand, the choice of controls matters for obtaining accurate estimates of $A_*, B_*$ (e.g. if we constantly play $u_t = 0$, then we get no information about $B_*$). Thus, the disturbances can be accurately estimated without accurately estimating $A_*, B_*$.

## 3.3 Running time

It suffices to argue that for each epoch $r$, a 2-approximate affine barycentric spanner of $\mathcal{M}_r$ can be computed in polynomial time. First, the conditions of Theorem 8 are satisfied: 1) $\mathcal{M}_r$ is compact, since $c$ is Lipschitz and so $C(\cdot \mid \widehat{A}_{t_r}, \widehat{B}_{t_r})$ is continuous, and 2) from the proof of Lemma 18 (Appendix B.1) we can see that not only $M_* \in \mathcal{M}_r$, but also there exists a small ball around $M_*$ which is contained in $\mathcal{M}_r$, so $\mathcal{M}_r$ is not contained in any proper affine subspace. Thus, to prove polynomial time, it suffices to have a linear optimization oracle for $\mathcal{M}_r$. Observe that $\mathcal{M}_r$ is convex, being an intersection of sublevel sets of convex functions. So, given a separation oracle for $\mathcal{M}_r$, a linear optimization oracle can be implemented in polynomial time via the ellipsoid method. Now, such a separation oracle can be implemented, given access to $C(M \mid \widehat{A}_{t_{r'}}, \widehat{B}_{t_{r'}})$ and $\nabla_M C(M \mid \widehat{A}_{t_{r'}}, \widehat{B}_{t_{r'}})$ for all $M \in \mathcal{M}$ and epochs $r' < r$. Even though we don't have exact access to these quantities because of the expectations they involve, we can approximate them in polynomial time up to $1/\mathrm{poly}(T)$-error by averaging samples, since we have offline access to the values $c(x, u)$ and gradients $\nabla_{x,u} c(x, u)$, for any $x, u$. Folklore approximation arguments suffice to show that even with this $1/\mathrm{poly}(T)$-error, ellipsoid method can optimize linear functions up to $1.01$ multiplicative error, in polynomial time. Now, just by inspecting the algorithm from [5] which is used to prove Theorem 8, we can see that even with a $1.01$-approximate optimization oracle (instead of an exact one) the same proof goes through. Thus, a 2-approximate affine barycentric spanner of $\mathcal{M}_r$ can be constructed in polynomial time.

## 4 Bandit feedback

To tackle online control with bandit feedback, we use the stochastic bandit convex optimization (SBCO) algorithm of [3] as a black-box. Before we present our algorithm and the formal theorem statement, we briefly present the SBCO setting. In SBCO, $\mathcal{X}$ is a convex subset of $\mathbb{R}^d$ with diameter bounded by $D$, and $f : \mathcal{X} \to \mathbb{R}$ is an $L$-Lipschitz convex function on $\mathcal{X}$. The algorithm has access to $f$ via a noisy value oracle, i.e., it can query the value of any $x \in \mathcal{X}$, and the response is $y = f(x) + \zeta$ where $\zeta$ is an independent $\sigma^2$-subgaussian random variable with mean zero. The goal is to minimize regret: after making $n$ queries $x_1, \ldots, x_n \in \mathcal{X}$, the regret is $\sum_{t=1}^{n} f(x_t) - n f(x_*)$, where $x_*$ is a minimizer of $f$ over $\mathcal{X}$. In [3], the authors give a polynomial-time algorithm that takes as input $d, D, L, \sigma^2, n$ and a separation oracle for $\mathcal{X}$, and achieves regret $\widetilde{O}(1) \cdot \mathrm{poly}(d, \log D, L, \sigma^2) \cdot \sqrt{n}$.

We now formally state our theorem, which says that after appropriately initializing the input parameters of the SBCO algorithm, Algorithm 4 achieves $\sqrt{T}$-regret.

**Theorem 17.** *There exist* $C_1, C_2, C_3, C_4, C_5 = poly\left(d_x, d_u, \kappa, \beta, \gamma^{-1}, \log T\right)$, *such that after initializing the SBCO algorithm with* $d = d_x \cdot d_u \cdot H$, $D = C_1$, $L = C_2$, $\sigma^2 = C_3$ *and* $n = T/(2H+2)$, *if the horizon* $T \geq C_4$, *then with high probability, warm-up exploration (Algorithm 6 in Appendix F) followed by Algorithm 4 satisfy* $R_T \leq C_5 \cdot \sqrt{T}$.

Our point here is that $\sqrt{T}$-regret is achievable in polynomial time, so we did not try to optimize the terms $C_i$. The proof is in Appendix C. The key step in the proof is showing that the SBCO algorithm of [3] is robust to adversarial noise in the responses, when this noise is small on average. This noise comes from $\|\widehat{w}_t - w_t\|$, which is small on average since we can show that Lemma 16 holds here too.

**Algorithm 4:** Control of unknown LDS with bandit feedback

---

**Input**: SBCO algorithm with parameters $d, D, L, \sigma^2, n$, domain $\mathcal{X} = \mathcal{M}$. Initial estimates
  $A_0, B_0$ satisfying $\|(A_0\ B_0) - (A_*\ B_*)\|_F \leq \epsilon$.
Set estimates of matrices $(\widehat{A}_1\ \widehat{B}_1) = (A_0\ B_0)$, noise estimates $\widehat{w}_t = 0$ for $t \leq 0$.
SBCO algorithm queries the first point $M$.
Set initial policy $M_1 = M$.
**for** $t = 1, \ldots, T$ **do**
     Apply control $u_t = \sum_{i=1}^{H} M_t^{[i-1]}\widehat{w}_{t-i}$.
     Observe $x_{t+1}$ and $c(x_t, u_t)$.
     Call System-Estimation (Algorithm 3), to get $\widehat{A}_{t+1}, \widehat{B}_{t+1}$.
     Record the estimate $\widehat{w}_t = x_{t+1} - \widehat{A}_{t+1}x_t - \widehat{B}_{t+1}u_t$.
     **if** $t \mod (2H + 1) = 0$ **then**
         Send $c(x_t, u_t)$ to SBCO algorithm.
         SBCO algorithm queries a new point $M$.
         Set $M_{t+1} = M$.
     **else**
         Set $M_{t+1} = M_t$.
     **end**
**end**

---

## 5  Extensions

**Linear Dynamical Controllers:**  Clearly, our regret bounds hold against any policy class that can be approximated by $\mathcal{M}$. One such class are all stabilizing linear-dynamical-controllers (for background on LDCs see [29]).

**General stochastic disturbances:**  Other than Gaussian, we can deal with any stochastic bounded disturbance distribution. The only place where the assumption of Gaussian disturbances really helps is that given some policy $M$ and matrices $\widehat{A}, \widehat{B}$, we can compute offline (to a very good approximation) the surrogate cost $C(M \mid \widehat{A}, \widehat{B})$, because we know the disturbance distribution. However, even when we don't, we can still use the estimated disturbances $(\widehat{w}_t)_t$ as samples to approximate this expectation (i.e., the average cost).

**Partial observation:**  The extension to partial observation is tedious but straightforward and uses the idea of "nature's y's", exactly as in [29].

## 6  Summary and open questions

We gave the first polynomial-time algorithms with optimal regret, with respect to the time horizon, for online control of LDS with general convex costs and comparator class the set of strongly-stable linear policies. Our main result was a novel geometric exploration scheme for the case where the cost function is known. The following open questions arise. First, can we improve the $\widetilde{O}(C) \cdot d_x d_u (d_x + d_u)\sqrt{T}$ regret bound, in terms of dimension dependence? This looks plausible because the barycentric spanners are constructed by treating the policies as flattened vectors of dimension $d_x d_u H$, thus the matrix structure is not exploited. Second, Algorithm 1 is not practical, since it employs the ellipsoid method. Is there a simpler, gradient-based algorithm that also achieves $\sqrt{T}$-regret? Third, a challenging question is whether $\sqrt{T}$-regret is achievable for nonstochastic control, where the disturbances are adversarial and the cost function adersarially changes over time. Even more broadly, can we prove regret bounds with respect to interesting nonlinear, yet tractable policy classes?

## 7  Broader Impact

This work does not present any foreseeable societal consequence.

## 8 Acknowledgements

EH acknowledges the support of NSF grant # 1704860 and Google.

## Footnotes

[1] In Section 5, we explain how we can extend all our results to hold for the more general class of stabilizing linear-dynamical-control policies.

[2]For unstable systems, without Assumption 4, the regret is exponential in the $n$ (see [10]).

[3]$\widetilde{O}(1)$ hides logarithmic factors.

[4]We can easily account for more general $L$-Lipschitz costs via rescaling. Also, we can account for quadratic costs, by assuming Lipschitzness inside a ball where state and control belong with high probability.

[5]We set $x_s = 0$, for $s \leq 0$.

[6]The reason we need the affine variant of barycentric spanners is technical, and due to the fact that $\Psi_i(M \mid A_*, B_*)$ are affine functions of $M$ (instead of just linear).

[7] On average the error is very small, because of the lower bound on $T$ in Theorem 9.

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
