[Supplementary Material]

## Additional notation

For two matrices $A, B$ we write $\langle A, B \rangle$ to denote the matrix inner product $tr(A^T B)$. We write $|A|$ to denote the determinant $det(A)$. Finally, for $x, y \in \mathbb{R}$ which depend on the problem parameters we write $x \lesssim y$ to denote that $x \leq O(1) \cdot y$.

## A  Initial stabilizing policy

**Assumption 4.** *The learner is initially given a policy $K_0 \in \mathcal{K}$.*

We can replace Assumption 1 with Assumption 4 and get all our results, by making a small change in our algorithms. The change is that instead of playing the control $u_t$ suggested by Algorithms 1,4,5, we play the control $\widetilde{u}_t = K_0 x_t + u_t$. We now explain why the same regret bounds for known cost function and bandit feedback hold for this case too. First, the LDS evolves as $x_{t+1} = \widetilde{A}_* x_t + B_* \widetilde{u}_t + w_t$, where $\widetilde{A}_* = A_* + B_* K_0$. Assumption 4 implies that $\widetilde{A}_*$ is $(\kappa, \gamma)$-strongly stable. Since Corollary 10 and Theorem 4 hold under Assumption 1, we can bound the regret of our algorithm with respect to the system $(\widetilde{A}_* \ B_*)$ and the class of $(\kappa, \gamma)$-strongly stable linear policies for that system. We call this policy class $\mathcal{K}(\widetilde{A}_*, B_*)$, which is defined in the same way as $\mathcal{K}$, after substituting $A_*$ with $\widetilde{A}_*$. Formally, the regret we described is defined as

$$\widetilde{R}_T^{st} = \sum_{t=1}^{T} c(x_t, u_t) - T \cdot \min_{K \in \mathcal{K}(\widetilde{A}_*, B_*)} \widetilde{J}(K), \tag{15}$$

where $\widetilde{J}(K) = \lim_{T \to \infty} \frac{1}{T} \mathbb{E}_{K, \widetilde{A}_*, B_*} \left[ \sum_{t=1}^{T} c(x_t, u_t) \right]$ and $\mathbb{E}_{K, \widetilde{A}_*, B_*}$ denotes expectation with respect to the system $(\widetilde{A}_* \ B_*)$ and policy $K$. The fact that we can get the same regret bounds under Assumption 4, follows from the observation that for all $K \in \mathcal{K}$, $K - K_0 \in \mathcal{K}(\widetilde{A}_*, B_*)$ and $J(K) = \widetilde{J}(K - K_0)$.

## B  Proofs for known cost function

Theorem 9 follows from Lemmas 11 and 12. We formally prove Lemma 12 in Appendix B.1 and Lemma 11 in Appendix B.5. For notational convenience, we define $M_{r,j} = M_{r,0}$, for $j = d + 1, d + 2, \ldots, 2d$. In our analysis, we substitute the "Call Execute-Policy($M_{r,0}, d \cdot T_r$)" with "for $j = d + 1, d + 2, \ldots 2d$ do Execute-Policy($M_{r,j}, T_r$) end." Clearly, these are equivalent, but the latter will lead to simpler formulas.

### B.1  Proof of Lemma 12

To bound $R_T^{st}$, we bound the suboptimality gap, i.e. $R^{st}(M) = \mathcal{C}(M \mid A_*, B_*) - \mathcal{C}(M_* \mid A_*, B_*)$, for all policies in $\mathcal{M}_{r+1}$.

**Lemma 18.** *With high probability, for all epochs $r$, we have*

- $M_* \in \mathcal{M}_r$, *and*
- *for all $M \in \mathcal{M}_{r+1}$, $R^{st}(M) \leq 5 \cdot 2^{-r}$.*

Given Lemma 18, we can conclude the proof of Lemma 12, as we show below.

*Proof.* $R_T^{st} = \sum_{t=1}^{T} R^{st}(M_t) = \sum_{r=1}^{q} \sum_{j=0}^{d} T_r \cdot R^{st}(M_{r,j})$, where $q$ is the total number of epochs. From Definition 7 of 2-approximate affine barycentric spanners, we have that $M_{r,j} \in \mathcal{M}_r$. Thus, Lemma 18 implies that with high probability, for all $r \geq 2$ and for all $j$, we have $R^{st}(M_{r,j}) \leq 5 \cdot 2^{-(r-1)}$. We will now bound $\sum_{r=1}^{q} 2^r$. Observe that $T_r = D \cdot 2^{2r}$, where $D = \widetilde{\Theta}(\kappa^4 \gamma^{-3}) \cdot d_x d_u (d_x + d_u)^2$, and $T \gtrsim \sum_{r=1}^{q} d \cdot T_r = D \cdot d \sum_{r=1}^{q} 2^{2r}$. Thus,

$\sqrt{\frac{T}{Dd}} \gtrsim \sqrt{\sum_{r=1}^{q} 2^{2r}} \geq q^{-1/2} \sum_{r=1}^{q} 2^r$, by Cauchy-Schwarz. Using that $q \leq O(\log T)$, we get $\sum_{r=1}^{q} 2^r \leq \widetilde{O}(1) \cdot \sqrt{\frac{T}{Dd}}$. Summarizing, by excluding the first epoch, we have

$$R_T^{st} - \sum_{j=1}^{2d} T_1 \cdot R^{st}(M_{j,1}) \leq \sum_{r=2}^{q} \sum_{j=0}^{d} D \cdot 2^{2r} \cdot 5 \cdot 2^{-(r-1)} \lesssim D \cdot d \cdot \sum_{r=1}^{q} 2^r \leq \widetilde{O}(1) \cdot \sqrt{D \cdot d \cdot T}. \tag{16}$$

In Appendix 66, we show that for all $M \in \mathcal{M}$, we have $R^{st}(M) \leq \widetilde{O}(\kappa^5 \beta^2 \gamma^{-2}) \cdot \sqrt{d_x}$. Thus,

$$R_T^{st} \leq 2d \cdot 2^2 \cdot D \cdot \widetilde{O}(\kappa^5 \beta^2 \gamma^{-2}) \cdot \sqrt{d_x} + \widetilde{O}(1) \cdot \sqrt{D \cdot d \cdot T}. \tag{17}$$

Since $T = \widetilde{\Omega}\left(\kappa^{14} \beta^4 \gamma^{-8}\right) \cdot d_x d_u (d_x + d_u)^5$, by substituting $D = \widetilde{\Theta}(\kappa^4 \gamma^{-3}) \cdot d_x d_u (d_x + d_u)^2$ and $d = \widetilde{O}(\gamma^{-1}) \cdot d_x d_u$, we get that $R_T^{st} \leq \widetilde{O}\left(\kappa^2 \gamma^{-2}\right) \cdot d_x d_u (d_x + d_u) \sqrt{T}$. $\qquad\square$

To prove Lemma 18, we prove Lemma 13 (stated in the main text), which ensures that for all policies $M \in \mathcal{M}_r$, the estimated surrogate cost $\mathcal{C}(M \mid \widehat{A}_{t_r}, \widehat{B}_{t_r})$ is close to the true surrogate cost $\mathcal{C}(M \mid A_*, B_*)$. Given Lemma 13, we can conclude the proof of Lemma 12, as we show below.

*Proof.* We condition on the event that for all $r$, $M \in \mathcal{M}_r$,

$$\left| \mathcal{C}(M \mid \widehat{A}_{t_r}, \widehat{B}_{t_r}) - \mathcal{C}(M \mid A_*, B_*) \right| \leq 2^{-r}. \tag{18}$$

For the first bullet of the lemma, suppose that for some $r$, $M_* \in \mathcal{M}_r$ and $M_* \notin \mathcal{M}_{r+1}$. Thus, there exists $M \in \mathcal{M}_r$, such that $\mathcal{C}(M \mid \widehat{A}_{t_r}, \widehat{B}_{t_r}) < \mathcal{C}(M_* \mid \widehat{A}_{t_r}, \widehat{B}_{t_r}) - 3\epsilon_r$. Then, inequality 18 implies that $\mathcal{C}(M \mid A_*, B_*) - \epsilon_r < \mathcal{C}(M_* \mid A_*, B_*) + \epsilon_r - 3\epsilon_r$, which contradicts the optimality of $M_*$.

For the second bullet, if $M \in \mathcal{M}_{r+1}$, then $\mathcal{C}(M \mid \widehat{A}_{t_r}, \widehat{B}_{t_r}) - \mathcal{C}(M_* \mid \widehat{A}_{t_r}, \widehat{B}_{t_r}) \leq 3\epsilon_r$, since we showed that $M_* \in \mathcal{M}_r$. By applying inequality 18, we get $\mathcal{C}(M \mid A_*, B_*) - \mathcal{C}(M_* \mid A_*, B_*) \leq 5\epsilon_r$. $\qquad\square$

It remains to prove Lemma 13. This lemma will follow from the following two lemmas and claim, where we use the covariance matrices $\Sigma(M)$, defined in Subsection 3.2.

**Lemma 19.** *Let $\widehat{A}, \widehat{B}$ be estimates of $A_*, B_*$, $\Delta = (\widehat{A}\ \widehat{B}) - (A_*\ B_*)$ and $\|\Delta\| \leq \frac{\gamma}{2\kappa^2}$. For all $M \in \mathcal{M}$, we have*

$$\left| \mathcal{C}(M \mid \widehat{A}, \widehat{B}) - \mathcal{C}(M \mid A_*, B_*) \right| \leq 6\kappa^2 \gamma^{-1} \left( \|\Delta^T\|_{\Sigma(M)} + 1/T \right). \tag{19}$$

We can apply this lemma for all $\Delta_t := (A_t\ B_t) - (A_*\ B_*)$, because of the following claim proved in Appendix G.

**Claim 20.** *With high probability, for all $t$, we have $\|\Delta_t\| \leq \gamma/(2\kappa^2)$.*

**Lemma 21.** *With high probability, we have $\|\Delta_{t_r}^T\|_{\Sigma(M)} \leq 2^{-r} \gamma/(12\kappa^2)$, for all epochs $r$ and $M \in \mathcal{M}_r$.*

Given these two lemmas and the claim, it is straightforward to get Lemma 13. Indeed, we get that with high probability, for all epochs $r$ and $M \in \mathcal{M}_r$,

$$\left| \mathcal{C}(M \mid \widehat{A}, \widehat{B}) - \mathcal{C}(M \mid A_*, B_*) \right| \leq 2^{-r}/2 + 6\kappa^2 \gamma^{-1}/T. \tag{20}$$

Since the length of the epochs increases exponentially, we have $r = O(\log(T))$. This, combined with the assumed lower bound on $T$ in the theorem statement, gives $6\kappa^2 \gamma^{-1}/T \leq 2^{-r}/2$, and we are done. The proof of Lemma 19 is in Appendix B.2 and the proof of Lemma 21 is in Appendix B.3.

## B.2 Proof of Lemma 19

We fix a policy $M \in \mathcal{M}$.

$$\left| \mathcal{C}(M \mid \widehat{A}, \widehat{B}) - \mathcal{C}(M \mid A_*, B_*) \right|$$

$$= \left| \mathbb{E}_\eta \left[ c\Big( x(M \mid \widehat{A}, \widehat{B}, \eta), \; u(M \mid \eta) \Big) \right] - \mathbb{E}_\eta \left[ c\Big( x(M \mid A_*, B_*, \eta), \; u(M \mid \eta) \Big) \right] \right|$$

$$\leq \; \mathbb{E}_\eta \left| c\Big( x(M \mid \widehat{A}, \widehat{B}, \eta), \; u(M \mid \eta) \Big) - c\Big( x(M \mid A_*, B_*, \eta), \; u(M \mid \eta) \Big) \right|$$

$$\leq \; \mathbb{E}_\eta \left\| x(M \mid \widehat{A}, \widehat{B}, \eta) - x(M \mid A_*, B_*, \eta) \right\|, \tag{21}$$

where we used the fact that $c$ is 1-Lipschitz.

To bound 21, we create two coupled dynamical systems: $x_1^{(1)} = x_1^{(2)} = 0$,

$$x_{t+1}^{(1)} = \widehat{A}x_t^{(1)} + \widehat{B}u_t + w_t \quad \text{and} \quad x_{t+1}^{(2)} = A_*x_t^{(2)} + B_*u_t + w_t, \tag{22}$$

where $w_t \overset{\text{i.i.d}}{\sim} N(0, I)$ and $u_t = \sum_{i=1}^{H} M^{[i-1]}w_{t-i}$ ($w_t = 0$ for $t \leq 0$). Observe that the coupling comes from the shared controls and disturbances. Let $z_t^{(1)} = \begin{pmatrix} x_t^{(1)} \\ u_t \end{pmatrix}$ and $z_t^{(2)} = \begin{pmatrix} x_t^{(2)} \\ u_t \end{pmatrix}$. We prove the following claim.

**Claim 22.** *The matrix $\widehat{A}$ is $(\kappa, \gamma/2)$-strongly stable* [8]*. Furthermore, for all $t \geq 2H + 2$, we have*

$$\left| \mathbb{E}_w \left\| x_t^{(1)} - x_t^{(2)} \right\| - \mathbb{E}_\eta \left\| x(M \mid \widehat{A}, \widehat{B}, \eta) - x(M \mid A_*, B_*, \eta) \right\| \right| \leq 1/T, \tag{23}$$

*where $w$ denotes the disturbance sequence $w_1, w_2 \ldots$*

*Proof.* We use the assumption that $\|\Delta\| \leq \frac{\gamma}{2\kappa^2}$, which implies that $\|\widehat{A} - A_*\| \leq \frac{\gamma}{2\kappa^2}$. Also, from Assumption 1, we have $A_* = Q\Lambda Q^{-1}$ with $\|\Lambda\| \leq 1 - \gamma$, and $\|Q\|, \|Q^{-1}\| \leq \kappa$. So, we get

$$\widehat{A} = Q\Lambda Q^{-1} + \widehat{A} - A_* = Q\left(\Lambda + Q^{-1}\left(\widehat{A} - A_*\right)Q\right)Q^{-1}. \tag{24}$$

Also, $\left\| \Lambda + Q^{-1}\left(\widehat{A} - A_*\right)Q \right\| \leq 1 - \gamma + \kappa^2\gamma/(2\kappa^2) = 1 - \gamma/2$. Thus, we proved that $\widehat{A}$ is $(\kappa, \gamma/2)$-strongly stable. Now, we prove 23:

$$\mathbb{E}_w \left\| x_t^{(1)} - x_t^{(2)} \right\|$$

$$= \mathbb{E}_w \left\| \widehat{A}^{H+1} x_{t-H-1}^{(1)} + \sum_{i=1}^{2H+1} \Psi_i(M \mid \widehat{A}, \widehat{B})w_{t-i} - A_*^{H+1} x_{t-H-1}^{(2)} - \sum_{i=1}^{2H+1} \Psi_i(M \mid A_*, B_*)w_{t-i} \right\|$$

$$\leq \left\| \widehat{A}^{H+1} \right\| \cdot \mathbb{E}_w \left\| x_{t-H-1}^{(1)} \right\| + \|A_*^{H+1}\| \cdot \mathbb{E}_w \left\| x_{t-H-1}^{(2)} \right\|$$

$$+ \mathbb{E}_w \left\| \sum_{i=1}^{2H+1} \Psi_i(M \mid \widehat{A}, \widehat{B})w_{t-i} - \sum_{i=1}^{2H+1} \Psi_i(M \mid A_*, B_*)w_{t-i} \right\|. \tag{25}$$

Since $t \geq 2H + 2$, the third term is exactly $\mathbb{E}_\eta \left\| x(M \mid \widehat{A}, \widehat{B}, \eta) - x(M \mid A_*, B_*, \eta) \right\|$. Now, we show that the first two terms are small. Since $A_*$ is $(\kappa, \gamma)$-strongly stable and $\widehat{A}$ is $(\kappa, \gamma/2)$-strongly stable, we have $\|A_*^{H+1}\| \leq \kappa^2(1-\gamma)^{H+1}$ and $\left\| \widehat{A}^{H+1} \right\| \leq \kappa^2(1-\gamma/2)^{H+1}$, where $H$ is defined in 2.1.1. In Appendix 56, we show that $\mathbb{E}_w \left\| x_{t-H-1}^{(1)} \right\|, \mathbb{E}_w \left\| x_{t-H-1}^{(2)} \right\| \leq O(\kappa^5\beta^2\gamma^{-2}) \cdot \sqrt{d_x}$. The way we chose $H$ finishes the proof. $\qquad\square$

Now, we fix a $t \geq 2H + 2$, whose exact value we choose later. We will bound $\mathbb{E}_w \left\| x_t^{(1)} - x_t^{(2)} \right\|$. First, we write a recursive formula for $x_t^{(1)} - x_t^{(2)}$:

$$\begin{aligned}
x_t^{(1)} - x_t^{(2)} &= \widehat{A} x_{t-1}^{(1)} + \widehat{B} u_{t-1} - A_* x_{t-1}^{(2)} - B_* u_{t-1} \\
&= (\widehat{A} - A_*) x_{t-1}^{(2)} + (\widehat{B} - B_*) u_{t-1} + \widehat{A}(x_{t-1}^{(1)} - x_{t-1}^{(2)}) \\
&= \Delta \cdot z_{t-1}^{(2)} + \widehat{A}(x_{t-1}^{(1)} - x_{t-1}^{(2)}).
\end{aligned} \tag{26}$$

By repeating 26, we get

$$x_t^{(1)} - x_t^{(2)} = \sum_{i=0}^{H-1} \widehat{A}^i \cdot \Delta \cdot z_{t-i-1}^{(2)} + \widehat{A}^H \left( x_{t-H-1}^{(1)} - x_{t-H-1}^{(2)} \right). \tag{27}$$

We prove a claim that shows that the second term is negligible.

**Claim 23.** $\mathbb{E}_w \left\| \widehat{A}^H \left( x_{t-H-1}^{(1)} - x_{t-H-1}^{(2)} \right) \right\| \leq 1/T$.

*Proof.* From Claim 22, we have that $\widehat{A}$ is $(\kappa, \gamma/2)$-strongly stable. Also, in Appendix 56, we show that $\mathbb{E}_w \left\| x_{t-H-1}^{(1)} \right\|, \mathbb{E}_w \left\| x_{t-H-1}^{(2)} \right\| \leq O(\kappa^5 \beta^2 \gamma^{-2}) \cdot \sqrt{d_x}$. Thus,

$$\begin{aligned}
\mathbb{E}_w \left\| \widehat{A}^H \left( x_{t-H-1}^{(1)} - x_{t-H-1}^{(2)} \right) \right\| &\leq \|\widehat{A}^H\| \cdot O(\kappa^5 \beta^2 \gamma^{-2}) \cdot \sqrt{d_x} \\
&\leq \kappa^2 (1-\gamma)^{H+1} \cdot O(\kappa^5 \beta^2 \gamma^{-2}) \cdot \sqrt{d_x}.
\end{aligned} \tag{28}$$

The way we chose $H$ finishes the proof. $\square$

By applying triangle inequality, we get

$$\mathbb{E}_w \| x_t^{(1)} - x_t^{(2)} \| \leq \sum_{i=0}^{H-1} \left\| \widehat{A}^i \right\| \cdot \mathbb{E}_w \left\| \Delta \cdot z_{t-i-1}^{(2)} \right\| + 1/T. \tag{29}$$

Now, we prove a claim which shows that for large $t$, the term $\mathbb{E}_w \left\| \Delta \cdot z_{t-i-1}^{(2)} \right\|$ is essentially time-independent, for all $i \in \{0, 1, \ldots, H-1\}$.

**Claim 24.** *For all $s \geq 2H + 2$, we have*

$$\left| \mathbb{E}_w \left\| \Delta \cdot z_s^{(2)} \right\| - \mathbb{E}_\eta \| \Delta \cdot z(M \mid A_*, B_*, \eta) \| \right| \leq 1/T. \tag{30}$$

*Proof.* It suffices to show that $\mathbb{E}_w \left\| \Delta \left( z_s^{(2)} - z(M \mid A_*, B_*, \eta(w)) \right) \right\| \leq 1/T$, where we define $\eta(w) := (w_{s-1}, w_{s-2}, \ldots, w_{s-2H-1})$. We have

$$\begin{aligned}
\mathbb{E}_w \left\| \Delta \left( z_s^{(2)} - z(M \mid A_*, B_*, \eta(w)) \right) \right\| &\leq \mathbb{E}_w \left\| z_s^{(2)} - z(M \mid A_*, B_*, \eta(w)) \right\| \\
&\leq \mathbb{E}_w \left\| x_s^{(2)} - x(M \mid A_*, B_*, \eta(w)) \right\| + \mathbb{E}_w \| u_s - u(M \mid \eta(w)) \| \\
&\leq \mathbb{E}_w \left\| A_*^{H+1} x_{s-H-1} \right\| + 0 \\
&\leq \left\| A_*^{H+1} \right\| \cdot \mathbb{E}_w \| x_{s-H-1} \|.
\end{aligned} \tag{31}$$

Since $A_*$ is $(\kappa, \gamma)$-strongly stable, Claim 56 and the way we chose $H$ finish the proof. $\square$

Now, we choose $t = 3H + 2$, which gives

$$\mathbb{E}_w \| x_t^{(1)} - x_t^{(2)} \| \leq (\mathbb{E}_\eta \| \Delta \cdot z(M \mid A_*, B_*, \eta) \| + 1/T) \cdot \sum_{i=0}^{H-1} \left\| \widehat{A}^i \right\| + 1/T. \tag{32}$$

Also, we have $\mathbb{E}_\eta \| \Delta \cdot z(M \mid A_*, B_*, \eta) \| \leq \left( \mathbb{E}_\eta \| \Delta \cdot z(M \mid A_*, B_*, \eta) \|^2 \right)^{1/2} = \left\| \Delta^T \right\|_{\Sigma(M)}$.

The following claim concludes the bound on $\mathbb{E}_w \| x_t^{(1)} - x_t^{(2)} \|$.

**Claim 25.** $\sum_{i=0}^{H-1} \left\| \widehat{A^i} \right\| \leq 4\kappa^2 \gamma^{-1}$.

*Proof.* This directly follows from the fact that $\widehat{A}$ is $(\kappa, \gamma/2)$-strongly stable (Claim 22). $\qquad\square$

After combining with Claim 22, we get

$$
\begin{aligned}
\mathbb{E}_\eta \left\| x(M \mid \widehat{A}, \widehat{B}, \eta) - x(M \mid A_*, B_*, \eta) \right\| &\leq 4\kappa^2 \gamma^{-1} \left( \left\| \Delta^T \right\|_{\Sigma(M)} + 1/T \right) + 2/T \\
&\leq 6\kappa^2 \gamma^{-1} \left( \left\| \Delta^T \right\|_{\Sigma(M)} + 1/T \right).
\end{aligned}
\tag{33}
$$

$\square$

## B.3 Proof of Lemma 21

We will prove the following two lemmas. The first controls the sum of squared errors $\|\Delta_{t_r}\|_{\Sigma(M_{r,j})}^2$, for the exploratory policies $M_{r,1}, \dots, M_{r,2d}$, and the second allows us to go from exploratory policies to all policies in $\mathcal{M}_r$.

**Lemma 26.** *With high probability,*

$$
\sum_{j=1}^{2d} \|\Delta_{t_r}^T\|_{\Sigma(M_{r,j})}^2 \leq 2^{-2r} \cdot \frac{\gamma^2}{12^2 \cdot 18 \cdot d\kappa^4} .
\tag{34}
$$

**Lemma 27.** *For all $M \in \mathcal{M}_r$, $\Sigma(M) \preccurlyeq 18d \cdot \sum_{j=0}^{d} \Sigma(M_{r,j})$.*

Given Lemmas 26 and 27, we conclude that with high probability, for all $M \in \mathcal{M}_r$, we have $\|\Delta_{t_r}^T\|_{\Sigma(M)}^2 \leq 18d \cdot \sum_{j=0}^{d} \|\Delta_{t_r}^T\|_{\Sigma(M_{r,j})}^2 \leq 2^{-2r}(\gamma/(12\kappa^2))^2$, and we are done. We prove Lemma 26 in B.3.1 and Lemma 27 in B.3.2.

### B.3.1 Proof of Lemma 26

To prove the lemma, we first prove the following two lemmas.

**Lemma 28.** *With high probability, for all $t$,*

$$
\left\| \Delta_t^T \right\|_{V_t} \leq \widetilde{O}(d_x + d_u),
\tag{35}
$$

*where $V_t = \sum_{s=1}^{t-1} z_s z_s^T + \lambda \cdot I$.*

**Lemma 29.** *With high probability, for all epochs $r$, we have $\sum_{j=1}^{2d} \Sigma(M_{r,j}) \preccurlyeq O(1/T_r) \cdot V_{t_r}$.*

Given these two lemmas, we have that with high probability

$$
\sum_{j=1}^{2d} \|\Delta_{t_r}^T\|_{\Sigma(M_{r,j})}^2 \leq O(1) \cdot \frac{\|\Delta_{t_r}^T\|_{V_{t_r}}^2}{T_r} \leq \widetilde{O}(1) \cdot \frac{(d_x + d_u)^2}{T_r} .
\tag{36}
$$

Substituting our choice of $T_r$ gives Lemma 26. We prove Lemma 28 in Appendix E. We now prove Lemma 29.

*Proof.* For $j = 1, 2, \dots, 2d$, let $I_{r,j} \subseteq [T]$ be the interval of execution of $M_{r,j}$ and $t_{r,j} \in [T]$ be the first step of this interval. Let $H' := 2H + 1$. For all $h = 0, 1, \dots, H'$, we define

$$
I_{r,j,h} = \{t \in I_{r,j} \mid t = t_{r,j} + H' \cdot k + h, \ k \geq 1\}.
\tag{37}
$$

We will show that with high probability, for all $r, j, h$, we have

$$
|I_{r,j,h}| \cdot \Sigma(M_{r,j}) \preccurlyeq O(1) \cdot \sum_{t \in I_{r,j,h}} z_t z_t^T + \frac{\lambda}{2dH'} \cdot I.
\tag{38}
$$

Once we have 38, we can finish the proof of Lemma 29. Indeed, summing over all $j, h$ gives

$$
\sum_{j=1}^{2d}(|I_{r,j}| - H') \cdot \Sigma(M_{r,j}) \preccurlyeq O(1) \cdot \sum_{j=1}^{2d}\sum_{h=0}^{H'-1}\sum_{t\in I_{r,j,h}} z_t z_t^T + \lambda \cdot I
$$

$$
\preccurlyeq O(1) \cdot \sum_{t=1}^{t_r-1} z_s z_s^T + \lambda \cdot I \preccurlyeq O(1) \cdot V_{t_r}, \tag{39}
$$

where we used that $\sum_{h=0}^{H'-1}|I_{r,j,h}| = |I_{r,j}| - H'$. Now, since $|I_{r,j}| - H' = T_r - H' \geq T_r/2$, we get $\sum_{j=1}^{2d}\Sigma(M_{r,j}) \preccurlyeq O(1/T_r) \cdot V_{t_r}$.

We now prove 38. We consider an auxiliary state/control sequence $(\overline{x}_t, \overline{u}_t)_{t\in[T]}$, defined as

$$
\overline{x}_t = \sum_{i=0}^{H} A_*^i w_{t-i-1} + \sum_{i=0}^{H} A_*^i B_* \overline{u}_{t-i-1} \tag{40}
$$

where $\overline{u}_s = \sum_{i=1}^{H} M_s^{[i-1]} w_{s-i}$.[9] The differences with the actual sequence are 1) we truncated the time-horizon and 2) here the controls use the true disturbances. We also define $\overline{z}_t = \begin{pmatrix}\overline{x}_t \\ \overline{u}_t\end{pmatrix}$. We will prove the following two claims.

**Claim 30.** *With high probability, for all $t$,*

$$
\|\overline{z}_t - z_t\| \leq O(\kappa^5 \beta^2 \gamma^{-1}) \cdot \sum_{i=1}^{2H+1} \|\widehat{w}_{t-i} - w_{t-i}\| + 1/T. \tag{41}
$$

**Claim 31.** *With high probability, for all $r, j, h$ we have*

$$
|I_{r,j,h}| \cdot \Sigma(M_{r,j}) \preccurlyeq O(1) \cdot \sum_{t\in I_{r,j,h}} \overline{z}_t \overline{z}_t^T. \tag{42}
$$

We show how to prove 38 using these two claims and then we prove them. Let $e_t = z_t - \overline{z}_t$ and $p = \kappa^5 \beta^2 \gamma^{-1}$. We condition on the event that the bounds of the two claims and of Lemma 35 hold. We have

$$
\|e_t\|^2 \leq 2/T^2 + O(p^2)(2H+1)\sum_{i=1}^{2H+1}\|w_{t-i} - \widehat{w}_{t-i}\|^2, \tag{43}
$$

where we used Claim 30 and Cauchy-Schwarz. We now fix a triple $(r, j, h)$. Summing over all $t \in I_{r,j,h}$,

$$
\sum_{t\in I_{r,j,h}}\|e_t\|^2 \leq \frac{2}{T} + O(p^2 H) \cdot \sum_{t\in I_{r,j,h}}\sum_{i=1}^{2H+1}\|w_{t-i} - \widehat{w}_{t-i}\|^2
$$

$$
\leq \frac{2}{T} + O(p^2 H) \cdot \sum_{t=1}^{T}\|w_t - \widehat{w}_t\|^2, \tag{44}
$$

where we used the definition of $I_{r,j,h}$. Lemma 35 implies that with high probability, for all $r, j, h$,

$$
\sum_{t\in I_{r,j,h}}\|e_t\|^2 \leq \widetilde{O}(\kappa^{10}\beta^4\gamma^{-3}) \cdot (d_x + d_u)^3. \tag{45}
$$

Moreover, $\overline{z}_t \overline{z}_t^T = (z_t - e_t)(z_t - e_t)^T \preccurlyeq 2z_t z_t^T + 2e_t e_t^T$, so

$$
\sum_{t\in I_{r,j,h}} \overline{z}_t \overline{z}_t^T \preccurlyeq 2\sum_{t\in I_{r,j,h}} z_t z_t^T + 2\sum_{t\in I_{r,j,h}} e_t e_t^T \preccurlyeq 2\sum_{t\in I_{r,j,h}} z_t z_t^T + 2\left(\sum_{t\in I_{r,j,h}}\|e_t\|^2\right) \cdot I, \tag{46}
$$

where we used that $\|\sum_{t\in I_{r,j,h}} e_t e_t^T\| \leq \sum_{t\in I_{r,j,h}} \|e_t e_t^T\| = \sum_{t\in I_{r,j,h}} \|e_t\|^2$. Now, Claim 31 and inequalities 45 and 46 give that with high probability, for all $r,j,h$,

$$|I_{r,j,h}| \cdot \Sigma(M_{r,j}) \preccurlyeq O(1) \cdot \left( 2 \left( \sum_{t\in I_{r,j,h}} \|e_t\|^2 \right) \cdot I + 2 \sum_{t\in I_{r,j,h}} z_t z_t^T \right)$$

$$\preccurlyeq \widetilde{O}(\kappa^{10}\beta^4\gamma^{-3})(d_x + d_u)^3 \cdot I + O(1) \cdot \sum_{t\in I_{r,j,h}} z_t z_t^T$$

$$\preccurlyeq \frac{\lambda}{H'(d+1)} \cdot I + O(1) \cdot \sum_{t\in I_{r,j,h}} z_t z_t^T, \tag{47}$$

where we used that $\lambda = \widetilde{\Theta}(\kappa^{10}\beta^4\gamma^{-5}) \cdot d_x d_u (d_x + d_u)^3$. It remains to prove Claims 30 and 31. We prove Claim 30 in Appendix 67. Now, we prove Claim 31. We use the following lemma, which is Theorem 1.1 of [18].

**Lemma 32** ([18]). *There exist positive constants $c_1, c_2$, such as the following hold. Let $\Sigma \in \mathbb{R}^{m\times m}$ positive semidefinite and $z_1, \ldots, z_n$ independent random vectors, distributed as $N(0, \Sigma)$. Let $\widehat{\Sigma} = 1/n \cdot \sum_{i=1}^n z_i z_i^T$. For all $\delta > 0$, there exists $c_3 = polylog(m, 1/\delta)$, such that if $n \geq c_3 \cdot m$, then with probability at least $1 - \delta$, $range(\widehat{\Sigma}) = range(\Sigma)$ and $\|\Sigma^{1/2}\widehat{\Sigma}^\dagger\Sigma^{1/2}\| \leq c_2$. [10]*

We can immediately get the following corollary.

**Corollary 33.** *For the setting of Lemma 32, with probability at least $1 - \delta$, we have $\Sigma \preccurlyeq O(1) \cdot \widehat{\Sigma}$.*

*Proof.* We have $\bar{\sigma}_{min}((\Sigma^\dagger)^{1/2}\widehat{\Sigma}(\Sigma^\dagger)^{1/2}) \geq 1/\Omega(1)$, where $\bar{\sigma}_{min}$ denotes the minimum nonzero singular value. Let $P$ be the projection matrix on $range(\Sigma)$. We have

$$\widehat{\Sigma} = P\widehat{\Sigma}P = \Sigma^{1/2}(\Sigma^\dagger)^{1/2}\widehat{\Sigma}(\Sigma^\dagger)^{1/2}\Sigma^{1/2}. \tag{48}$$

So, for all $x \in \mathbb{R}^m$, we have

$$x^T\widehat{\Sigma}x = x^T\Sigma^{1/2}(\Sigma^\dagger)^{1/2}\widehat{\Sigma}(\Sigma^\dagger)^{1/2}\Sigma^{1/2}x \geq \bar{\sigma}_{min}((\Sigma^\dagger)^{1/2}\widehat{\Sigma}(\Sigma^\dagger)^{1/2})x^T\Sigma x. \tag{49}$$

$\square$

Now, we apply Corollary 33 to show Claim 31. We fix a triple $(r, j, h)$. Notice that $M_{r,j}$ is a random variable that depends only on the disturbances that took place up to epoch $r - 1$. On the other hand, the random vectors $(\bar{z}_t)_{t\in I_{r,j,h}}$ are independent of each other and independent of all the disturbances that took place up to epoch $r - 1$, which follows from the definitions of $I_{r,j,h}$ and of the sequence $(\bar{z}_t)_t$. Thus, we have $\mathbb{E}_w\left[\bar{z}_t\bar{z}_t^T \mid M_{r,j}\right] = \Sigma(M_{r,j})$, where $w$ denotes the sequence $(w_t)_t$. So, after conditioning on $M_{r,j}$, we can apply Corollary 33 with $\Sigma = \Sigma(M_{r,j})$ and the set of vectors being $(\bar{z}_t)_{t\in I_{r,j,h}}$. Since $|I_{r,j,h}| = T_r/H'$ and $T_r$ is chosen to be large enough, we get $|I_{r,j,h}| \cdot \Sigma(M_{r,j}) \preccurlyeq O(1) \cdot \sum_{t\in I_{r,j,h}} \bar{z}_t\bar{z}_t^T$. Since, $(\bar{z}_t)_{t\in I_{r,j,h}}$ are independent of $M_{r,j}$, we get that with high probability, $|I_{r,j,h}| \cdot \Sigma(M_{r,j}) \preccurlyeq O(1) \cdot \sum_{t\in I_{r,j,h}} \bar{z}_t\bar{z}_t^T$. This was for a fixed $r, j, h$, so union bound concludes the proof. $\square$

The above proof also concluded the proof of 38. $\square$

### B.3.2 Proof of Lemma 27

We fix an $M \in \mathcal{M}_r$. Since $\{M_{r,0}, \ldots, M_{r,d}\}$ is a 2-approximate affine barycentric spanner of $\mathcal{M}_r$, we can write $M = M_{r,0} + \sum_{j=0}^d \lambda_j \cdot (M_{r,j} - M_{r,0})$, where $\lambda_j \in [-2, 2]$. Since we defined $M_{r,j} = M_{r,0}$ for all $j \in \{d+1, \ldots, 2d\}$, we can write $M = \sum_{j=1}^{2d} \lambda_j \cdot M_{r,j}$, where $\lambda_j = -\lambda_{j-d} + 1/d$ for all $j \geq d + 1$ (the other $\lambda_j$ stay the same). Thus, we have that all $\lambda_j \in [-3, 3]$ and $\sum_{j=1}^{2d} \lambda_j = 1$, i.e, it is an affine combination. The next claim, takes us from policies to covariances.

**Claim 34.** *The exists an affine transformation $T$, such that for all $M \in \mathcal{M}$, $\Sigma(M) = T(M)T(M)^T$.*

*Proof.* $\Sigma(M) = \mathbb{E}_\eta \left[ z(M \mid A_*, B_*, \eta) \cdot z(M \mid A_*, B_*, \eta)^T \right]$. For this proof, we write $\Psi(M)$, to denote $\Psi(M \mid A_*, B_*)$. We define

$$T(M) := \begin{pmatrix} \Psi_0(M) & \Psi_1(M) & \cdots & \Psi_{H-1}(M) & \Psi_H(M) & \cdots & \Psi_{2H}(M) \\ M^{[0]} & M^{[1]} & & M^{[H-1]} & 0 & \cdots & 0 \end{pmatrix} \quad (50)$$

and observe that $z(M \mid A_*, B_*, \eta) = T(M) \cdot \eta$. The transformation $T(\cdot)$ is affine due to the definition of $\Psi_i(M)$ (see Subsection 2.1.1). The claim follows from the fact that $\mathbb{E}_\eta \left[ \eta \eta^T \right] = I$. $\qquad\square$

Back to our fixed $M \in \mathcal{M}_r$, Claim 34 implies that $T(M) = \sum_{j=0}^d \lambda_j T(M_{r,j})$. We have

$$\Sigma(M) = T(M)T(M)^T = \left( \sum_{j=1}^{2d} \lambda_j T(M_{r,j}) \right) \left( \sum_{j=1}^{2d} \lambda_j T(M_{r,j}) \right)^T$$

$$\preccurlyeq \left( \sum_{j=1}^{2d} \lambda_j^2 \right) \left( \sum_{j=1}^{2d} T(M_{r,j}) T(M_{r,j})^T \right) \preccurlyeq 18d \cdot \sum_{j=1}^{2d} \Sigma(M_{r,j}), \quad (51)$$

where we used a generalized Cauchy-Schwartz that we prove in Appendix 57. $\qquad\square$

### B.4 Estimating the disturbances

We prove a lemma which guarantees that the disturbance estimates $\widehat{w}_t$ are on average very accurate.

**Lemma 35.** *With high probability, we have $\sum_{t=1}^T \|\widehat{w}_t - w_t\|^2 \leq \widetilde{O}(1) \cdot (d_x + d_u)^3$.*

*Proof.* After playing action $u_t$ and then observing $x_{t+1}$, we record the estimates $\widehat{A}_{t+1}$, $\widehat{B}_{t+1}$ and $\widehat{w}_t = x_{t+1} - \widehat{A}_{t+1}x_t - \widehat{B}_{t+1}u_t$. Let $V_0 = \lambda \cdot I$, where $\lambda$ is defined in Algorithm 3, and $V_t = V_0 + \sum_{s=1}^{t-1} z_s z_s^T$. Also, we remind the reader that $\Delta_t = (A_t \; B_t) - (A_* \; B_*)$. We have

$$\|\widehat{w}_t - w_t\|^2 = \left\| x_{t+1} - \widehat{A}_{t+1}x_t - \widehat{B}_{t+1}u_t - (x_{t+1} - A_*x_t - B_*u_t) \right\|^2$$

$$= \|\Delta_{t+1}z_t\|^2 \leq \left\| \Delta_{t+1} V_{t+1}^{\frac{1}{2}} \right\|^2 \cdot \left\| V_{t+1}^{-\frac{1}{2}} z_t \right\|^2.$$

Lemma 28 implies that with high probability $\left\| \Delta_{t+1} V_{t+1}^{\frac{1}{2}} \right\|^2 \leq \widetilde{O}(1) \cdot (d_x + d_u)^2$, for all $t$. The bound on $\sum_{t=1}^T \left\| V_{t+1}^{-\frac{1}{2}} z_t \right\|^2 = \sum_{t=1}^T \|z_t\|_{V_{t+1}^{-1}}^2$ follows from a linear-algebraic inequality, which has previously appeared in the context of online optimization [16] and stochastic linear bandits [13]. We state this inequality below.

**Lemma 36** ([20]). *Let $V_0$ positive definite and $V_t = V_0 + \sum_{s=1}^{t-1} x_s x_s^T$, where $x_1, \ldots, x_T \in \mathbb{R}^n$ is a sequence of vectors with $\|x_t\| \leq L$, for all $t$. Then*

$$\sum_{t=1}^T \left( 1 \wedge \|x_t\|_{V_{t+1}^{-1}}^2 \right) \leq 2n \log \left( \frac{tr(V_0) + TL^2}{n \det^{1/n}(V_0)} \right). \quad (52)$$

We get $\sum_{t=1}^T \left( 1 \wedge \|z_t\|_{V_{t+1}^{-1}}^2 \right) \leq \widetilde{O}(d_x + d_u)$, by applying this lemma for our sequence $z_1, \ldots, z_T$. The following claim completes the proof.

**Claim 37.** *With high probability, for all $t$, we have $\|z_t\|_{V_{t+1}^{-1}}^2 \leq 1$.*

*Proof.* From Claim 60 we have that with high probability, $\|z_t\| \leq \widetilde{O}(\kappa^5 \beta^2 \gamma^{-2})\sqrt{d_x}$, for all $t$. On the other hand, $\lambda I \preccurlyeq V_{t+1}$. Our choice of $\lambda$ (Algorithm 3) finishes the proof. $\qquad\square$

$\square$

**Remark 38.** It is worth noting that the proof of Lemma 35 does not use any information about the way we choose the controls. On the other hand, the choice of controls matters for obtaining accurate estimates of $A_*, B_*$. Thus, the disturbances can be accurately estimated without accurately estimating $A_*, B_*$.

## B.5 Proof of Lemma 11

In the proof, we use $\mathcal{C}(M)$ to refer to $\mathcal{C}(M|A_*, B_*)$.

$$R_T - R_T^{st} = \sum_{t=1}^{T} (c(x_t, u_t) - \mathcal{C}(M_t)) + T \min_{M \in \mathcal{M}} \mathcal{C}(M) - T \min_{K \in \mathcal{K}} J(K) \tag{53}$$

$$\leq \sum_{t=1}^{T} (c(x_t, u_t) - \mathcal{C}(M_t)) + 1, \tag{54}$$

where we used Theorem 5. We proceed with some definitions. We use the intervals $I_{r,j}$ and $I_{r,j,h}$ that we defined in the beginning of proof of Lemma 29. Also, let $I'_{r,j} = \{t_{r,j}, t_{r,j} + 1, \dots, t_{r,j} + H' - 1\}$, i.e. the first $H'$ steps of $I_{r,j}$. Observe that $\cup_{h=0}^{H'-1} I_{r,j,h} = I_{r,j} \setminus I'_{r,j}$. Let $q$ be the total number of epochs.

$$\sum_{t=1}^{T} (c(x_t, u_t) - \mathcal{C}(M_t)) = \sum_{r=1}^{q} \sum_{j=1}^{2d} \sum_{t \in I_{r,j}} (c(x_t, u_t) - \mathcal{C}(M_t))$$

$$= \sum_{r=1}^{q} \sum_{j=1}^{2d} \sum_{h=0}^{H'-1} \sum_{t \in I_{r,j,h}} (c(x_t, u_t) - \mathcal{C}(M_t)) + \sum_{r=1}^{q} \sum_{j=1}^{2d} \sum_{t \in I'_{r,j}} (c(x_t, u_t) - \mathcal{C}(M_t)) \tag{55}$$

We bound these two terms via the following two claims.

**Claim 39.** *With high probability,*

$$\sum_{r=1}^{q} \sum_{j=1}^{2d} \sum_{h=0}^{H'-1} \sum_{t \in I_{r,j,h}} (c(x_t, u_t) - \mathcal{C}(M_t)) \leq \widetilde{O}(\kappa^5 \beta^2 \gamma^{-2.5}) \cdot \sqrt{(d_x + d_u)^3 T}. \tag{56}$$

**Claim 40.** *With high probability,*

$$\sum_{r=1}^{q} \sum_{j=1}^{2d} \sum_{t \in I'_{r,j}} (c(x_t, u_t) - \mathcal{C}(M_t)) \leq \widetilde{O}(\kappa^5 \beta^2 \gamma^{-4}) \cdot d_x^{3/2} d_u. \tag{57}$$

These two claims and the fact that we have assumed $T \geq \gamma^{-2}$ conclude the proof of Lemma 11. We first prove Claim 40.

*Proof.* We will use the fact that the number of policy switches is small, i.e. logarithmic in $T$. First, we will need the following claim, which we prove in Appendix 67.

**Claim 41.** *With high probability, for all t,*

$$c(x_t, u_t) - \mathcal{C}(M_t) \leq \widetilde{O}(\kappa^5 \beta^2 \gamma^{-2}) \cdot \sqrt{d_x}. \tag{58}$$

Using Claim 41,

$$\sum_{r=1}^{q} \sum_{j=1}^{2d} \sum_{t \in I'_{r,j}} (c(x_t, u_t) - \mathcal{C}(M_t)) \leq q \cdot 2d \cdot 3H \cdot \widetilde{O}(\kappa^5 \beta^2 \gamma^{-2}) \cdot \sqrt{d_x}$$

$$\leq \widetilde{O}(\kappa^5 \beta^2 \gamma^{-4}.) \cdot d_x^{3/2} d_u. \tag{59}$$

$\square$

Now, we prove Claim 39.

*Proof.* We break the sum into two terms: the first will be controlled via martingale concentration and the second will be errors coming from truncation of horizon-type arguments and the fact that the algorithm uses $\widehat{w}_t$ instead of $w_t$. We use the auxiliary sequence $(\bar{z}_t)_t$ defined in Appendix B.3.1. From Claim 67, we have that with high probability, for all $t$,

$$\|\bar{z}_t - z_t\| \leq \widetilde{O}(\kappa^5\beta^2\gamma^{-1}) \cdot \sum_{i=1}^{2H+1} \|\widehat{w}_{t-i} - w_{t-i}\| + 1/T. \tag{60}$$

So, we write

$$\sum_{r=1}^{q}\sum_{j=1}^{2d}\sum_{h=0}^{H'-1}\sum_{t\in I_{r,j,h}} (c(x_t, u_t) - \mathcal{C}(M_t))$$

$$= \sum_{h=0}^{H'-1}\sum_{r=1}^{q}\sum_{j=1}^{2d}\sum_{t\in I_{r,j,h}} (c(\bar{z}_t) - \mathcal{C}(M_{r,j})) + \sum_{r=1}^{q}\sum_{j=1}^{2d}\sum_{h=0}^{H'-1}\sum_{t\in I_{r,j,h}} (c(z_t) - c(\bar{z}_t)) \tag{61}$$

For the second sum, we have that with high probability,

$$\sum_{r=1}^{q}\sum_{j=1}^{2d}\sum_{h=0}^{H'-1}\sum_{t\in I_{r,j,h}} (c(z_t) - c(\bar{z}_t)) \leq \sum_{t=1}^{T}\|z_t - \bar{z}_t\| \leq \widetilde{O}(\kappa^5\beta^2\gamma^{-1})\sum_{t=1}^{T}\sum_{i=1}^{2H+1}\|\widehat{w}_{t-i} - w_{t-i}\| + 1$$

$$\leq \widetilde{O}(\kappa^5\beta^2\gamma^{-2})\sum_{t=1}^{T}\|\widehat{w}_t - w_t\| + 1, \tag{62}$$

where we used inequality 60 and the fact that $c$ is 1-Lipschitz. We now apply Lemma 35, followed by Cauchy-Schwartz, to get $\sum_{t=1}^{T}\|\widehat{w}_t - w_t\| \leq \widetilde{O}(1) \cdot \sqrt{(d_x + d_u)^3 T}$. Thus, we showed that the second sum in 62 is at most $\widetilde{O}(\kappa^5\beta^2\gamma^{-2}) \cdot \sqrt{(d_x + d_u)^3 T}$.

We will now bound the first sum, i.e., $\sum_{h=0}^{H'-1}\sum_{r=1}^{q}\sum_{j=1}^{2d}\sum_{t\in I_{r,j,h}} (c(\bar{z}_t) - \mathcal{C}(M_{r,j}))$. We will bound each

$$\sum_{r=1}^{q}\sum_{j=1}^{2d}\sum_{t\in I_{r,j,h}} (c(\bar{z}_t) - \mathcal{C}(M_{r,j})) \tag{63}$$

separately. We define the $\sigma$-algebra $\mathcal{F}_t = \sigma(w_1, w_2, \ldots, w_{t-H'-1})$. We also fix a tuple $(h, r, j, t)$, where $h \in \{0, 1, \ldots, H'\}$, $r \in \{1, 2, \ldots, q\}$, $j \in \{1, 2, \ldots, 2d\}$, and $t \in I_{r,j,h}$. Now, we focus on the sum in 63 which corresponds to our fixed $h$. Observe that for all $s < t$, if $c(\bar{z}_s) - \mathcal{C}(M_s)$[11] participates in this sum, then it is $\mathcal{F}_t$-measurable. This is because of 1) the way the algorithm decides which policies to execute and 2) the definition of the sequence $(\bar{z}_t)_t$. Moreover, the policy $M_{r,j}$ is also $\mathcal{F}_t$-measurable, because at time $t$ we have already spent at least $H'$ timesteps in epoch $r$, so everything that happened up until the end of epoch $r-1$ is $\mathcal{F}_t$-measurable, and so the same is true for $M_{r,j}$. Combining these observations with the definitions of $\bar{z}_t$ and $\mathcal{C}(M_{r,j})$, we get that $\mathbb{E}[c(\bar{z}_t) - \mathcal{C}(M_{r,j})|\mathcal{F}_t] = 0$. To apply martingale concentration we will need the following claim.

**Claim 42.** *There exists $\sigma = O(\kappa^5\beta^2\gamma^{-2})$, such that $\bar{z}_t$ is $\sigma$-Lipschitz as a function of $(w_{t-2H-1}, \ldots, w_{t-1})$. Furthermore, conditioned on $\mathcal{F}_t$, the random variable $c(\bar{z}_t) - \mathcal{C}(M_{r,j})$ is $\sigma^2$-subgaussian.*

*Proof.* Since $M_{r,j} \in \mathcal{M}$, for all $s \in \{t - H - 1, t - H, \ldots, t\}$, $\bar{u}_s$ is $L_u = \sum_{i=1}^{H}\|M_{r,j}^{[i-1]}\| = O(\kappa^3\beta\gamma^{-1})$-Lipschitz. Thus, the Lipschitz constant of $\bar{x}_t$ is upper-bounded by

$$\sum_{i=0}^{H}\|A_*^i\| + \sum_{i=0}^{H}\|A_*^i\| \cdot \|B_*\| \cdot L_u \leq O(\kappa^2\gamma^{-1} + \kappa^2\gamma^{-1}\beta \cdot L_u), \tag{64}$$

where we used Claim 54. Substituting $L_u$ finishes the proof of the first part. The second part follows from the fact that $c$ is 1-Lipschitz and from Gaussian concentration [32]. □

From Azuma's inequality [31], for our fixed $h$, the random variable in 63 is $\sigma^2 \frac{T}{3H}$-subgaussian, with $\sigma$ as in Claim 42, and so with high probability it is at most $O\left(\sqrt{\sigma^2 T/H}\right)$. By applying union bound, we get that with high probability the first sum in 73 is at most $O\left(\sqrt{\sigma^2 TH}\right) = \widetilde{O}(\kappa^5 \beta^2 \gamma^{-2.5}) \cdot \sqrt{T}$. □

□

## C   Bandit feedback: Proof of Theorem 17

We will prove the following theorem.

**Theorem 43.** *There exist $C_1, C_2, C_3, C_4, C_5 = \mathrm{poly}\left(d_x, d_u, \kappa, \beta, \gamma^{-1}, \log T\right)$, such that after initializing the SBCO algorithm with $d = d_x \cdot d_u \cdot H$, $D = C_1$, $L = C_2$, $\sigma^2 = C_3$ and $n = T/(2H+2)$, the following holds. If $T \geq C_4$, the intial state $\|x_1\| \leq \widetilde{O}(\kappa^2 \beta \gamma^{-1/2}) \cdot \sqrt{d_x}$, and the initial estimation error bound $\|(A_0\ B_0) - (A_*\ B_*)\|_F \leq \epsilon$ satisfies $\epsilon^2 \leq (C_6 \cdot d_x d_u (d_x + d_u))^{-1}$, where $C_6 = \kappa^{10} \beta^4 \gamma^{-5}$, then with high probability, Algorithm 4 satisfies $R_T \leq C_5 \cdot \sqrt{T}$.*

Given the above theorem, Theorem 17 follows from the analysis of warmup exploration given in Appendix F (specifically Lemma 63). Theorem 43 follows from the following two lemmas (similarly to the case of known cost function).

**Lemma 44.** *With high probability, $R_T - R_T^{st} \leq \mathrm{poly}\left(d_x, d_u, \kappa, \beta, \gamma^{-1}, \log T\right) \cdot \sqrt{T}$.*

**Lemma 45.** *With high probability, $R_T^{st} \leq \mathrm{poly}\left(d_x, d_u, \kappa, \beta, \gamma^{-1}, \log T\right) \cdot \sqrt{T}$.*

To prove these lemmas, we will first need a bound for $\sum_{t=1}^{T} \|\widehat{w}_t - w_t\|^2$.

**Lemma 46.** *With high probability, Algorithm 4 satisfies $\sum_{t=1}^{T} \|\widehat{w}_t - w_t\|^2 \leq \widetilde{O}(1) \cdot (d_x + d_u)^3$.*

The proof of the lemma is exactly the same with the proof of Lemma 35, so we do not repeat it here. Second, we require a generalization of the SBCO setup (Appendix C.1). After this, we prove Lemma 44 in Appendix C.2 and Lemma 45 in Appendix C.3.

### C.1   SBCO: robustness to small adversarial perturbations and low number of swtiches

We consider a small generalization of the SBCO setup, where the learner observes the function values under the sum of a stochastic and a small (on average) adversarial corruption. We will show that we can properly set the hyperparameters of the SBCO algorithm from [3], to get $\sqrt{n}$ regret efficiently ($n$ is the time horizon), in this more general setting. We will also note some useful properties of that algorithm and we will finally give some preliminaries related to its application in Algorithm 4.

**Setting**

Let $\mathcal{X}$ be a convex subset of $\mathbb{R}^d$, for which we have a separation oracle and has diameter bounded by $D$. Let $f : \mathcal{X} \to \mathbb{R}$ be an $L$-Lipschitz convex function on $\mathcal{X}$. We have noisy black-box access to $f$. Specifically, we are allowed to do $n$ queries: at time $t$ we query $x_t$ and the response is

$$y_t = f(x_t) + \zeta_t + \xi_t \tag{65}$$

where $\zeta_t$ conditioned on $(\zeta_1, \ldots, \zeta_{t-1})$ is $\sigma_\zeta^2$-subgaussian with mean 0 [12]. The sequence $\xi_1, \ldots, \xi_n$ can be completely adversarial and can even depend on $\{\zeta_t\}_{t \in [n]}$. However, the magnitude of this

adversarial noise satisfies the following constraint: with probability at least $1 - 1/n^c$,

$$\sum_{t=1}^{n} \xi_t^2 \leq \sigma_\xi^2, \tag{66}$$

for some parameters $c$ [13], $\sigma_\xi \geq 0$. The algorithm incurs a cost $f(x_t)$ for the query $x_t$. The goal is to minimize regret:

$$\sum_{t=1}^{n} \left( f(x_t) - f(x_*) \right), \tag{67}$$

where $x_*$ is a minimizer of $f$ over $\mathcal{X}$. Clearly, the standard SBCO setting [3] is recovered when $\xi_t = 0$ for all $t$. The algorithm in [3] uses a hyperparameter $\sigma$, which is set to be $\sigma_\zeta$. We will show that for this more general setting that we described, we can get the same regret guarantee (up to a factor depending on $\sigma_\xi$), by setting $\sigma := \sqrt{c+1} \cdot \max(\sigma_\zeta, \sigma_\xi)$ and running the same algorithm.

**Regret bound**

**Theorem 47.** *With probability at least $1 - O(n^{-c})$, the algorithm in [3] (page 11) initialized with hyperparameter $\sigma = \sqrt{c+1} \cdot \max(\sigma_\zeta, \sigma_\xi)$ has regret*

$$\sum_{t=1}^{T} \left( f(x_t) - f(x_*) \right) \leq \text{poly} \left( \sigma, d, L, \log n, \log D \right) \cdot \sqrt{n}. \tag{68}$$

*Proof.* Every time this algorithm queries a new point $x$, it queries it multiple times and takes the average of the responses to reduce the variance. More specifically, the algorithm maintains a parameter $\gamma$ which is the desired estimation accuracy. If at time $t$, the point to be queried is new (different than the one at time $t - 1$), then it queries it $s = 4 \cdot \frac{\sigma^2}{\gamma^2} \log n$ times[14]) and receives $y_t, \ldots, y_{t+s-1}$. Then, the algorithm computes the average $avg_t = 1/s \cdot \sum_{i=0}^{s-1} y_{t+i}$. In [3], the proof of the regret bound (which is the same as the RHS of 68) uses the fact that the noise is stochastic only in order to argue that with probability at least $1 - \delta$, the error $|avg_t - f(x_t)| \leq \gamma$, for all $t$. Once they have this, their analysis implies that the regret bound holds with probability at least $1 - \delta$. The proof of our theorem is essentially that this condition also holds in our setting (for $\delta = 1 - O(n^{-c})$), if we set $\sigma = \sqrt{c+1} \cdot \max(\sigma_\zeta, \sigma_\xi)$. Indeed, we have

$$avg_t = f(x_t) + \frac{\sum_{i=0}^{s-1} \zeta_{t+i}}{s} + \frac{\sum_{i=0}^{s-1} \xi_{t+i}}{s}. \tag{69}$$

- Stochastic component: $s = 4 \cdot \frac{\sigma^2}{\gamma^2} \log n \geq (c+1) \cdot \frac{\sigma_\zeta^2}{(\gamma/2)^2} \log n$, so from Azuma's inequality: $\left| \frac{\sum_{i=0}^{s-1} \zeta_{t+i-1}}{s} \right| \leq \gamma/2$, with probability at least $1 - O(n^{-(c+1)})$. A union bound implies that the bound holds for all $t$, with probability at least $1 - O(n^{-c})$.

- Adversarial component: by applying Cauchy-Schwarz, we get that with probability at least $1 - O(n^{-c})$, for all $t$,

$$\left| \frac{\sum_{i=0}^{s-1} \xi_{t+i}}{s} \right| \leq \frac{\sqrt{s} \sqrt{\sum_{i=0}^{s-1} \xi_{t+i}^2}}{s} \leq \frac{\sigma_\xi}{\sqrt{s}} \leq \gamma/2. \tag{70}$$

$\square$

Other than the regret guarantee, we will also need some other properties of the SBCO algorithm. To present these, we need a high level description of this algorithm, which we now provide.

**High level description of the SBCO algorithm**

Let $H_t = (x_i, y_i)_{i=1}^t$ [15], i.e., the history up to time $t$. There exists a function $g$ that is polynomial-time computable and takes as input $H_t$ (for any $t$) and outputs a pair $(x, s)$, which indicates that the algorithm will query $x$ for the timesteps $t+1, t+2, \ldots, t+s$. More specifically, given this function $g$, the SBCO algorithm has the following form.

---

Set $t = 1$.
Set $r = 1$.
**while** $t \leq n$ **do**
    Set $j_r = t$ (switching time).
    Set $(x, s) = g(H_{t-1})$.
    Query $x$ for the timesteps $t, t+1, \ldots, t+s-1$.
    Set $t = t + s$.
    Set $r = r + 1$.
**end**

---

The way the function $g$ is constructed makes sure that the above algorithm queries exactly $n$ points. We now state two facts about this algorithm, the first follows from the above description and the second from inspecting the full algorithm (page 11 of [3]).

**Fact 48.** *If $t \in [j_r, j_{r+1})$, then $x_t$ (point queried at time $t$) is $\sigma(H_{j_r-1})$-measurable.*

**Fact 49.** *At the end of the algorithm, the index $r \leq poly(\log n, d, \sigma, \log D, L)$.*

Note that Fact 49 says that the number of point-switches is only logarithmic in $n$.

**SBCO algorithm in Algorithm 4: preliminaries**

Let $H' = 2H + 1$, $n = \lfloor T/H' \rfloor$ and $M(1), M(2), \ldots, M(n)$ be the points/policies queried by the SBCO algorithm in Algorithm 4. Observe that for all $t$, if $t = (j-1)H' + h$ for some $h \in \{1, \ldots, H'\}$, then the executed policy at time $t$ is $M_t = M(j)$. Also, let $j_1 \leq j_2 \leq \ldots \leq j_k$ be the switching timesteps of the SBCO algorithm (as in the high-level description of Section C.1). We define $t_r = (j_r - 1)H' + 1$ and $t_{k+1} = T + 1$. Observe that the executed policy $M_t$ remains constant for all $t \in [t_r, t_{r+1})$. Also, Fact 48 directly implies the following claim that we will use later.

**Claim 50.** *If $t \in [t_r, t_{r+1})$, then $M_t$ is $\sigma((x_s, u_s)_{s=1}^{t_r-1})$-measurable.*

**C.2   Proof of Lemma 44**

The proof is similar to the proof of Lemma 11. We use $\mathcal{C}(M)$ to denote $\mathcal{C}(M \mid A_*, B_*)$.

$$R_T - R_T^{st} = \sum_{t=1}^{T}(c(z_t) - \mathcal{C}(M_t)) + T \min_{M \in \mathcal{M}} \mathcal{C}(M) - T \min_{K \in \mathcal{K}} J(K)$$

$$\leq \sum_{t=1}^{T}(c(z_t) - \mathcal{C}(M_t)) + O(1), \tag{71}$$

where we used Theorems 5, 6. We proceed with some definitions. For all $r \in \{1, \ldots, k\}$, $h \in \{0, 1, \ldots, H'-1\}$, we define the intervals $I_r = [t_r, t_{r+1})$, $I_{r,h} = \{t \in I_r \mid t = t_r + H' \cdot j + h, j \geq 1\}$ and $I'_r = \{t_r, t_r + 1, \ldots, t_r + H' - 1\} = I_r \setminus (\cup_{h=0}^{H'-1} I_{r,h})$. We have

$$\sum_{t=1}^{T}(c(z_t) - \mathcal{C}(M_t)) = \sum_{r=1}^{k} \sum_{t \in I_r}(c(z_t) - \mathcal{C}(M_t))$$

$$= \sum_{r=1}^{k} \sum_{h=0}^{H'-1} \sum_{t \in I_{r,h}}(c(z_t) - \mathcal{C}(M_t)) + \sum_{r=1}^{k} \sum_{t \in I'_r}(c(z_t) - \mathcal{C}(M_t)) \tag{72}$$

We call the first sum $S_1$ and the second $S_2$. In Appendix G (Claim 69), we show that with high probability, $c(z_t) - \mathcal{C}(M_t) \leq \widetilde{O}(\kappa^5 \beta^2 \gamma^{-2}) \cdot \sqrt{d_x}$, for all $t$. Combining with Fact 49, we get that with high probability, $S_2 \leq k \cdot H' \cdot \widetilde{O}(\kappa^5 \beta^2 \gamma^{-2}) \cdot \sqrt{d_x} = \mathrm{poly}(d_x, d_u, \kappa, \beta, \gamma^{-1}, \log T)$. To bound $S_1$, we use the auxiliary sequence $(\bar{z}_t)_t$, defined in Appendix B.5.

$$S_1 = \sum_{h=0}^{H'-1} \sum_{r=1}^{k} \sum_{t \in I_{r,h}} (c(\bar{z}_t) - \mathcal{C}(M_t)) + \sum_{r=1}^{k} \sum_{h=0}^{H'-1} \sum_{t \in I_{r,h}} (c(z_t) - c(\bar{z}_t)) \qquad (73)$$

We call the first sum $S_3$ and the second $S_4$. We first bound $S_4$. In Appendix G (Claim 67), we show that with high probability, for all $t$,

$$\|\bar{z}_t - z_t\| \lesssim \kappa^5 \beta^2 \gamma^{-1} \cdot \sum_{i=1}^{2H+1} \|\widehat{w}_{t-i} - w_{t-i}\| + 1/T. \qquad (74)$$

So, we have that with high probability,

$$S_4 \leq \sum_{t=1}^{T} \|z_t - \bar{z}_t\| \leq \widetilde{O}(\kappa^5 \beta^2 \gamma^{-2}) \sum_{t=1}^{T} \|\widehat{w}_t - w_t\| + 1 \qquad (75)$$

We now apply Lemma 46, followed by Cauchy-Schwartz, to get that with high probability, $\sum_{t=1}^{T} \|\widehat{w}_t - w_t\| \leq \widetilde{O}(1) \cdot \sqrt{(d_x + d_u)^3 T}$. Thus, we showed that $S_4 \leq \mathrm{poly}(d_x, d_u, \kappa, \beta, \gamma^{-1}, \log T) \cdot \sqrt{T}$. The final step is to bound $S_3 = \sum_{h=0}^{H'-1} S_{3,h}$, where $S_{3,h} = \sum_{r=1}^{k} \sum_{t \in I_{r,h}} (c(\bar{z}_t) - \mathcal{C}(M_t))$. We will show that with high probability, for all $h$, $S_{3,h} \leq \mathrm{poly}(d_x, d_u, \kappa, \beta, \gamma^{-1}, \log T) \cdot \sqrt{T}$, which will conclude the proof. We will prove the following claim.

**Claim 51.** *Let $\mathcal{F}_t = \sigma(w_1, w_2, \ldots, w_{t-H'-1})$. Let $r \in \{1, \ldots, k\}$, $h \in \{0, \ldots, H'-1\}$, $t \in I_{r,h}$. The following hold.*

- *If $t' \leq t - H'$, then $c(\bar{z}_{t'}) - \mathcal{C}(M_{t'})$ is $\mathcal{F}_t$-measurable.*

- $\mathbb{E}[c(\bar{z}_t) - \mathcal{C}(M_t) \mid \mathcal{F}_t] = 0$.

- *Conditioned on $\mathcal{F}_t$, $c(\bar{z}_t) - \mathcal{C}(M_t)$ is $\mathrm{poly}(\kappa, \beta, \gamma^{-1}, \log T)$-subgaussian.*

Given this claim, we can apply Azuma's inequality and a union bound to bound $S_{3,h}$, for all $h$. It remains to prove the claim.

*Proof.* First, we show that if $t' \leq t$, then $M_{t'}$ is $\mathcal{F}_t$-measurable. Indeed, from Claim 50, we get that $M_{t'}$ is $\sigma((x_s, u_s)_{s=1}^{t_r - 1})$-measurable. Also, we have $\sigma((x_s, u_s)_{s=1}^{t_r - 1}) \subseteq \sigma(w_1, w_2, \ldots, w_{t_{r'}-2}) \subseteq \mathcal{F}_t$, since $t \geq t_r + H'$.

Now, we show the first bullet. Let $t' \leq t - H'$. Then, from the argument above, $M_{t'}$ is $\mathcal{F}_t$-measurable. Also, $\bar{z}_{t'}$ is $\sigma(w_1, w_2, \ldots, w_{t'-1})$-measurable and $\sigma(w_1, w_2, \ldots, w_{t'-1}) \subseteq \mathcal{F}_t$, since $t \geq t' + H'$.

For the second bullet, notice that conditioned on $\mathcal{F}_t$, the only source of randomness in $c(\bar{z}_t) - \mathcal{C}(M_t)$ are the $w_{t-H'}, \ldots, w_{t-1}$. Since $t \in I_{r,h}$, at time $t$ the policy $M_t$ has already been executed for the last $H'$ steps. Thus, $\mathbb{E}[c(\bar{z}_t) - \mathcal{C}(M_t) \mid \mathcal{F}_t] = 0$.

For the third bullet, it is easy to see that $\bar{z}_t$ is $\mathrm{poly}(\kappa, \beta, \gamma^{-1})$-Lipschitz as a function of $(w_{t-H'}, \ldots, w_{t-1})$. This, combined with gaussian concentration [32] completes the proof. $\qquad\square$

$\square$

## C.3  Proof of Lemma 45

We first prove the following lemma.

**Lemma 52.** *Under the conditions of Theorem 43, we have*

$$\sum_{j=1}^{n}(\mathcal{C}(M(j)) - \mathcal{C}(M_*)) \leq \mathrm{poly}(d_x, d_u, \kappa, \beta, \gamma^{-1}, \log T) \cdot \sqrt{n}. \tag{76}$$

Given this lemma, Lemma 45 immediately follows, since $R_T = H' \cdot \sum_{j=1}^{n}(\mathcal{C}(M(j)) - \mathcal{C}(M_*))$. We now give the proof of Lemma 52.

*Proof.* Clearly, there exist $C_1, C_2 \leq \mathrm{poly}(d_x, d_u, \kappa, \beta, \gamma^{-1}, \log T)$, such that $\mathcal{C}(M)$ is $C_1$-Lipschitz and the diameter of $\mathcal{M}$ is at most $C_2$. It suffices to show that when the SBCO algorithm queries $M(j) = M_t$, where $t = (j-1)H' + 1$, the response $c(z_{t+H'-1})$ can be written as

$$c(z_{t+H'-1}) = \mathcal{C}(M(j)) + \zeta(j) + \xi(j), \tag{77}$$

where

- conditioned on $\zeta(1), \ldots, \zeta(j-1)$, the noise $\zeta(j)$ is $\mathrm{poly}(d_x, d_u, \kappa, \beta, \gamma^{-1}, \log T)$-subgaussian, and

- with high probability, $\sum_{j=1}^{n} \xi(j)^2 \leq \mathrm{poly}(d_x, d_u, \kappa, \beta, \gamma^{-1}, \log T)$.

We will use the auxiliary sequence $(\overline{z}_t)_t$ defined in Appendix B.5, to write

$$c(z_{t+H'-1}) = \mathcal{C}(M_t) + (c(\overline{z}_{t+H'-1}) - \mathcal{C}(M_t)) + (c(z_{t+H'-1}) - c(\overline{z}_{t+H'-1})). \tag{78}$$

The second term is $\zeta(j)$ and the third is $\xi(j)$. The guarantee on $\zeta(j)$ follows from Claim 51. For the guarantee on $\sum_{j=1}^{n} \xi(j)^2$, we have $\sum_{j=1}^{n} \xi(j)^2 \leq \sum_{t=1}^{T} \|z_t - \overline{z}_t\|^2$. By Claim 67, we have $\sum_{t=1}^{T} \|z_t - \overline{z}_t\|^2 \leq O(1) + \mathrm{poly}(\kappa, \beta, \gamma^{-1}) \cdot \sum_{t=1}^{T} \|w_t - \widehat{w}_t\|^2$. Lemma 46 concludes the proof. $\square$

# D   Auxiliary claims

**Claim 53.** *With high probability, for all $t$,*

$$\|w_t\| \leq \widetilde{O}(\sqrt{d_x}). \tag{79}$$

*Proof.* This follows from standard concentration of the norm of gaussian random vectors [32]. $\square$

**Claim 54.** *For all $i \in \mathbb{N}$, $\|A_*^i\| \leq \kappa^2(1-\gamma)^i$.*

*Proof.* Using Assumption 1, we have $\|A_*^i\| = \|Q\Lambda^i Q^{-1}\| \leq \kappa^2(1-\gamma)^i$. $\square$

**Claim 55.** *There exists a $Z = \widetilde{O}(\kappa^5 \beta^2 \gamma^{-2}) \cdot \sqrt{d_x}$, such that the following hold. For any policy $M \in \mathcal{M}$, we have $\mathbb{E}_\eta \|z(M|A_*, B_*, \eta)\| \leq Z$. Furthermore, suppose that $\|x_1\| \leq \widetilde{O}(\kappa^2 \beta \gamma^{-1/2}) \cdot \sqrt{d_x}^{16}$, and that instead of executing our algorithms, we play $u_t = \sum_{i=1}^{H} M_t^{[i-1]} w_{t-i}$ for all $t$, where $(M_t)_t$ is an arbitrary policy sequence. Then, with high probability, we have $\|z_t\| \leq Z$.*

*Proof.* First, we fix a policy $M \in \mathcal{M}$. For this proof, we write $u(M \mid \eta_{i:i+H-1}) = \sum_{j=1}^{H} M^{[j-1]} \eta_{i+j-1}$, where $\eta_{i:i+H-1}$ denotes the sequence $\eta_i, \eta_{i+1}, \ldots, \eta_{i+H-1}$. So, we have $\mathbb{E}_\eta \|z(M \mid A_*, B_*, \eta)\| \leq \mathbb{E}_\eta \|x(M \mid A_*, B_*, \eta)\| + \mathbb{E}_\eta \|u(M \mid \eta_{0:H-1})\|$. Now, for all $i = \{0, 1, \ldots, H+1\}$, we have

$$\left(\mathbb{E}_\eta \|u(M \mid \eta_{i:i+H-1})\|\right)^2 \leq \mathbb{E}_\eta \|u(M \mid \eta_{i:i+H-1})\|^2 = tr\left(\sum_{j=1}^{H} \left(M^{[j-1]}\right)^T \cdot M^{[j-1]}\right)$$

$$\leq d_x \sum_{j=1}^{H} \left(\kappa^3 \beta(1-\gamma)^j\right)^2$$

$$\lesssim \kappa^6 \beta^2 \gamma^{-1} \cdot d_x. \tag{80}$$

Thus, we bounded $\mathbb{E}_\eta \| u(M \mid \eta_{0:H-1}) \| \lesssim \kappa^3 \beta \gamma^{-1/2} \cdot \sqrt{d_x}$. Now, we write

$$x(M \mid A_*, B_*\eta) = \sum_{i=0}^{H} A_*^i \cdot \eta_i + \sum_{i=0}^{H} A_*^i B_* u(M \mid \eta_{i+1:i+H}) \tag{81}$$

By triangle inequality,

$$\mathbb{E}_\eta \| x(M \mid A_*, B_*\eta) \| \leq \sum_{i=0}^{H} \| A_*^i \| \cdot \mathbb{E}_\eta \| \eta_i \| + \sum_{i=0}^{H} \| A_*^i \| \cdot \| B_* \| \cdot \mathbb{E}_\eta \| u(M \mid \eta_{i+1:i+H}) \|$$

$$\lesssim \sqrt{d_x} \kappa^2 \sum_{i=0}^{H} (1-\gamma)^i + \kappa^2 \beta \sum_{i=0}^{H} (1-\gamma)^i \cdot \kappa^3 \beta \gamma^{-1/2} \cdot \sqrt{d_x}, \tag{82}$$

where we used inequality 80. Thus, we get $\mathbb{E}_\eta \| x(M \mid A_*, B_*\eta) \| \lesssim \kappa^5 \beta^2 \gamma^{-3/2} \cdot \sqrt{d_x}$.

Now, we will bound $\|z_t\|$. First, we assumed that $\|x_1\| \leq \widetilde{O}(\beta \kappa^2 \gamma^{-1/2}) \cdot \sqrt{d_x}$. Also, the disturbance bound from Claim 53 and the spectral bounds on $M_t^{[i]}$, imply that with high probability, for all t, we have $\|u_t\| \leq \widetilde{O}(\kappa^3 \beta \gamma^{-1}) \cdot \sqrt{d_x}$. We now show that for large enough $Z = \widetilde{O}(\kappa^5 \beta^2 \gamma^{-2}) \cdot \sqrt{d_x}$, after conditioning on $\|x_1\| \leq \widetilde{O}(\beta \kappa^2 \gamma^{-1/2}) \cdot \sqrt{d_x}$ and $w_t \leq \widetilde{O}(\sqrt{d_x})$ and $\|u_t\| \leq \widetilde{O}(\kappa^3 \beta \gamma^{-1}) \cdot \sqrt{d_x}$ for all t, we have $\|x_t\| \leq Z/2$, for all t.

$$\|x_{t+1}\| \leq \| A_*^t x_1 \| + \sum_{i=0}^{t-1} \| A_*^i \| \cdot \| w_{t-i} \| + \sum_{i=0}^{t-1} \| A_*^i \| \cdot \| B_* \| \cdot \| u_{t-i} \| \tag{83}$$

Using the bounds on disturbances, controls and $\|x_1\|$, we get that $\|x_{t+1}\|$ is at most

$$\widetilde{O}(\beta \kappa^2 \gamma^{-1/2}) \cdot \sqrt{d_x} \cdot \kappa^2 + \sqrt{d_x} \cdot \kappa^2 \cdot \sum_{i=0}^{\infty} (1-\gamma)^i + \kappa^2 \beta \cdot \sum_{i=0}^{\infty} (1-\gamma)^i \cdot \widetilde{O}(\kappa^3 \beta \gamma^{-1}) \cdot \sqrt{d_x}, \tag{84}$$

which is at most $Z/2$. $\qquad\square$

**Claim 56.** For all $t \geq H+2$, $\mathbb{E}_w \left\| x_{t-H-1}^{(1)} \right\|, \mathbb{E}_w \left\| x_{t-H-1}^{(2)} \right\| \leq O(\kappa^5 \beta^2 \gamma^{-2}) \cdot \sqrt{d_x}$.

*Proof.* First, for all t, $\mathbb{E}_w \|u_t\| \leq \sum_{i=1}^{H} \| M^{[i-1]} \| \cdot \mathbb{E}_w \| w_{t-i} \| \leq \kappa^3 \beta \gamma^{-1} \cdot \sqrt{d_x}$. We have

$$x_{t-H-1}^{(1)} = \sum_{i=1}^{t-H-3} \widehat{A}^i w_{t-H-2-i} + \sum_{i=1}^{t-H-3} \widehat{A}^i \widehat{B} u_{t-H-2-i}. \tag{85}$$

Also, in Claim 22 we proved that $\widehat{A}$ is $(\kappa, \gamma/2)$-stronlgy stable. Also, we have $\| \widehat{B} - B_* \| \leq \gamma/(2\kappa^2)$, so $\| \widehat{B} \| \leq \beta + 1$. Thus, we get

$$\mathbb{E}_w \| x_{t-H-1}^{(1)} \| \leq \sum_{i=1}^{t-H-3} \| \widehat{A}^i \| \cdot \mathbb{E}_w \| w_{t-H-2-i} \| + \sum_{i=1}^{t-H-3} \| \widehat{A}^i \| \cdot \| \widehat{B} \| \cdot \mathbb{E}_w \| u_{t-H-2-i} \|$$

$$\leq \sqrt{d_x} \kappa^2 \sum_{i=0}^{\infty} (1-\gamma/2)^i + \kappa^2 (\beta+1) \sum_{i=0}^{\infty} (1-\gamma/2)^i \cdot \kappa^3 \beta \gamma^{-1} \cdot \sqrt{d_x}$$

$$\lesssim \kappa^5 \beta^2 \gamma^{-2} \cdot \sqrt{d_x}. \tag{86}$$

Since $A_*$ is $(\kappa, \gamma)$-stronlgy stable and $\| B_* \| \leq \beta$, the same calculation gives $\mathbb{E}_w \| x_{t-H-1}^{(2)} \| \lesssim \kappa^5 \beta^2 \gamma^{-2} \cdot \sqrt{d_x}$. $\qquad\square$

**Claim 57.** Let $\lambda_1, \ldots, \lambda_n \in \mathbb{R}$ and $A_1, \ldots, A_n$ matrices with compatible dimensions. Then,

$$\left( \sum_{j=1}^{n} \lambda_j A_j \right) \left( \sum_{j=1}^{n} \lambda_j A_j \right)^T \preccurlyeq \left( \sum_{j=1}^{n} \lambda_j^2 \right) \left( \sum_{j=1}^{n} A_j A_j^T \right) \tag{87}$$

*Proof.* Without loss of generality, it suffices to prove the result for the case where $A_j$ have each one column. Then, for all vectors $x$,

$$
x^T \left( \sum_{j=1}^n \lambda_j A_j \right) \left( \sum_{j=1}^n \lambda_j A_j \right)^T x = \left( \sum_{j=1}^n \lambda_j A_j^T x \right) \cdot \left( \sum_{j=1}^n \lambda_j A_j^T x \right)
$$
$$
\leq \left( \sum_{j=1}^n \lambda_j^2 \right) \left( \sum_{j=1}^n (A_j^T x)^2 \right)
$$
$$
= x^T \left( \sum_{j=1}^n \lambda_j^2 \right) \left( \sum_{j=1}^n A_j A_j^T \right) x. \tag{88}
$$

$\square$

# E  Proof of Lemma 28

We use $|Q|$ to denote the determinant of matrix $Q$. We first state Lemma 6 of [12].

**Lemma 58.** *Let $V_0 = \lambda \cdot I$. With high probability, for all $t$,*

$$
\|\Delta_t^T\|_{V_t}^2 \leq \widetilde{O}(1) \cdot (d_x + d_u) \log \frac{|V_t|}{|V_0|} + T_0 \|\Delta_0\|_F^2. \tag{89}
$$

Since $|V_0| = \lambda^{d_x + d_u}$, $\|\Delta_0\|_F \leq \epsilon \leq (d_x + d_u)$ (condition of Theorem 9) and $\lambda = T_0$, we get that with high probability, for all $t$,

$$
\|\Delta_t^T\|_{V_t}^2 \leq \widetilde{O}(1) \cdot (d_x + d_u)^2 \left( 1 + \log |V_t|^{(d_x + d_u)^{-1}} \right). \tag{90}
$$

The following claim helps control $|\overline{V}_t|$.

**Claim 59.** *For all $t$,*

$$
|V_t|^{(d_x + d_u)^{-1}} \leq \frac{1}{d_x + d_u} \cdot \sum_{s=1}^{t-1} \|z_s\|^2 + \lambda \sqrt{T}. \tag{91}
$$

*Proof.* From AMGM,

$$
|V_t|^{(d_x + d_u)^{-1}} \leq \frac{1}{d_x + d_u} \cdot \mathrm{tr}(V_t) = \frac{1}{d_x + d_u} \cdot \mathrm{tr} \left( \sum_{s=1}^{t-1} z_s z_s^T + \lambda \sqrt{T} \right)
$$
$$
= \frac{1}{d_x + d_u} \cdot \sum_{s=1}^{t-1} \|z_s\|^2 + \lambda \sqrt{T}.
$$

$\square$

So, it remains to control the magnitude of $z_t$.

**Lemma 60.** *Suppose $\|x_1\| \leq \widetilde{O}(\kappa^2 \beta \gamma^{-1/2}) \cdot \sqrt{d_x}$ [17]. Then, with high probability, both Algorithms 1 and 4 satisfy for all $t$,*

$$
\|z_t\| \leq \widetilde{O}(\kappa^5 \beta^2 \gamma^{-2}) \cdot \sqrt{d_x}. \tag{92}
$$

*Proof.* It suffices to show that if both 90 and 79 hold for all $t$, then 92 holds. We prove this by induction on $t$. For $t = 1$, we have $u_1 = 0$ (for both algorithms) and $\|x_1\|$ is bounded by assumption. Suppose that $z_s$ satisfies 92 for all $s < t$. Then, Claim 59 implies that for all $s < t$,

$$
\log |V_{s+1}|^{(d_x + d_u)^{-1}} \leq \widetilde{O}(1) \tag{93}
$$

and so by 90 we have

$$\left\|\Delta_{s+1}^T\right\|_{V_{s+1}}^2 \ \leq \ \widetilde{O}(1) \cdot (d_x + d_u)^2. \tag{94}$$

We now bound $u_s$ for $s \leq t$. Fix an $s \leq t$. We have

$$u_s = \sum_{i=1}^{H} M_s^{[i-1]} \widehat{w}_{s-i} = \sum_{i=1}^{H} M_s^{[i-1]} w_{s-i} + \sum_{i=1}^{H} M_s^{[i-1]} \left(\widehat{w}_{s-i} - w_{s-i}\right). \tag{95}$$

We show the following claim.

**Claim 61.** *For all $\tau < t$, we have $\|\widehat{w}_\tau - w_\tau\| \ \leq \ \widetilde{O}(1)$.*

*Proof.* We have

$$\|\widehat{w}_\tau - w_\tau\|^2 = \|\Delta_{\tau+1} \cdot z_\tau\|^2 \ \leq \ \widetilde{O}(\kappa^{10} \beta^4 \gamma^{-4}) \cdot d_x \cdot \|\Delta_{\tau+1}\|. \tag{96}$$

From inequality 94 and the fact that $\lambda \cdot I \preccurlyeq V_{s+1}$, we get that $\|\Delta_{\tau+1}\| \ \leq \ \widetilde{O}(1) \cdot (d_x + d_u)^2/T_0$. The claim follows from our choice of $\lambda$. $\square$

Back to our fixed $s \leq t$, using Claim 61 and Claim 53, we get

$$\|u_s\| \ \leq \ \widetilde{O}(1) \cdot \sqrt{d_x} \cdot \sum_{i=1}^{H} \|M_s^{[i-1]}\| \ \leq \ \widetilde{O}(\kappa^3 \beta \gamma^{-1}) \cdot \sqrt{d_x} . \tag{97}$$

We will now bound $\|x_t\|$.

$$\begin{aligned}
\|x_t\| &= \left\|A_*^{t-1} x_1 + \sum_{i=1}^{t-1} A_*^{i-1} \left(w_{t-i} + B_* u_{t-i}\right)\right\| \\
&\leq \ \|A_*^{t-1}\| \|x_1\| + \sum_{i=1}^{t-1} \|A_*^{i-1}\| \left(\|w_{t-i}\| + \|B_*\| \|u_{t-i}\|\right) \\
&\leq \ \widetilde{O}(\beta \kappa^2 \gamma^{-1/2}) \cdot \sqrt{d_x} \cdot \|A_*^{t-1}\| + \widetilde{O}(\kappa^3 \beta^2 \gamma^{-1}) \cdot \sum_{i=0}^{\infty} \|A_*^i\| \\
&\leq \ \widetilde{O}(\kappa^5 \beta^2 \gamma^{-2}) \cdot \sqrt{d_x},
\end{aligned}$$

where the last step follows from $(\kappa, \gamma)$-strong stability of $A_*$. The fact that $\|z_t\| \ \leq \ \|x_t\| + \|u_t\|$ finishes the proof. $\square$

We can now finish the proof of Lemma 28. With high probability, both 90 and 92 hold for all $t$, so $\log |\overline{V}_t|^{(d_x+d_u)^{-1}} \ \leq \ \widetilde{O}(1)$, which after being plugged-in inequality 90 completes the proof.

# F  Warm-up

---
**Algorithm 5:** Warm-up exploration
---
Set $T_0 = \lambda$, where $\lambda$ is defined in Algorithm 3.
**for** $t = 1, 2, \ldots, T_0$ **do**
    Observe $x_t$.
    Play $u_t \sim N(0, I)$.
**end**
Set $V = \sum_{t=1}^{T_0} z_t z_t^T + (\kappa^2 + \beta)^{-2} \cdot I$. [a]
Compute $(A_0 \ B_0) = \sum_{t=1}^{T_0} x_{t+1} z_t^T V^{-1}$.
---

> [a] $z_t$ is defined as in Algorithm 3.

To get the initial estimates $A_0, B_0$ we conduct the warm-up exploration given in Algorithm 6. In the main text we use $x_1$ to denote the state after the warm-up period (i.e., $x_{T_0+1}$). This "reset" of time is done for simplifying the presentation in the main text. From Theorem 20 and Appendix B.2 in [12], we automatically get the following lemma.

**Lemma 62.** *Let $\Delta_0 = (A_0\ B_0) - (A_*\ B_*)$. With high probability,*

$$\|\Delta_0\|_F^2 \ \leq\ \widetilde{O}(1) \cdot \frac{(d_x + d_u)^2}{T_0} = (C_1 d_x d_u (d_x + d_u))^{-1}, \tag{98}$$

*where $C_1 = \kappa^{10} \beta^4 \gamma^{-5}$.*

Now, we bound the regret overhead caused by the above process.

**Lemma 63.** *Let $C = \kappa^{13} \beta^5 \gamma^{-5.5}$. With high probability, the regret incurred during warm-up exploration is at most*

$$\tilde{O}(C) \cdot d_x^{3/2} d_u (d_x + d_u)^3 \ \leq\ \tilde{O}(C) \cdot d_x d_u (d_x + d_u) \sqrt{T}, \tag{99}$$

*and the state at the end of it, i.e., $x_{T_0+1}$ has norm $\|x_{T_0+1}\| \ \leq\ \widetilde{O}(\kappa^2 \beta \gamma^{-1/2}) \cdot \sqrt{d_x}$.*

*Proof.* We define the regret at step $t$ to be $c(z_t) - J(K_*)$, where $K_* \in \arg\min_{K \in \mathcal{K}} J(K)$. We prove the following claim.

**Claim 64.** *During warm-up period, the regret at step $t$ is at most $\|z_t\| + O(\kappa^3 \gamma^{-1/2} \sqrt{d_x})$.*

*Proof.* Let $K_* \in \arg\min_{K \in} J(K)$. For all $t$, we have

$$c(x_t, u_t) - J(K_*) = c(x_t, u_t) - \lim_{T \to \infty} \frac{1}{T} \cdot \mathbb{E}_{K_*} \sum_{t=1}^{T} c(x_t^{K_*}, u_t^{K_*})$$

$$= \lim_{T \to \infty} \frac{1}{T} \cdot \sum_{t=1}^{T} \mathbb{E}_{K_*} \left[ c(x_t, u_t) - c(x_t^{K_*}, u_t^{K_*}) \right]$$

$$\leq \lim_{T \to \infty} \frac{1}{T} \cdot \sum_{t=1}^{T} \mathbb{E}_{K_*} \left[ \|z_t\| + \|x_t^{K_*}\| + \|u_t^{K_*}\| \right]$$

$$\leq \|z_t\| + Q,$$

where $Q$ is such that $\mathbb{E}_{K_*}[\|x_t^{K_*}\| + \|u_t^{K_*}\|] \ \leq\ Q$, for all $t$. Claim 65 shows that $Q$ can be chosen to be at most $O(\kappa^3 \gamma^{-1/2} \sqrt{d_x})$, which finishes the proof. $\square$

Now we use that $\|z_t\| \ \leq\ \|x_t\| + \|u_t\|$, and we bound $\|x_t\|$ and $\|u_t\|$.

- With high probability, $\|u_t\| \ \leq\ \widetilde{O}(\sqrt{d_x})$, for all $t \in [T_0]$. Indeed, for $t \in [T_0]$ we have $u_t \sim N(0, I)$, so the bound on $\|u_t\|$ follows as in the proof of Claim 53.

- Now, we bound $\|x_t\|$. For all $t \in [T_0 + 1]$, $x_t \sim N(0, \Sigma)$, where

$$\Sigma = \sum_{i=0}^{t-2} A_*^i (I + B_* B_*^T) (A_*^T)^i \tag{100}$$

  From Claim 54, we have $\left\| (A_*^T)^i A_*^i \right\| \ \leq\ \|A_*^i\|^2 \ \leq\ \kappa^4 (1 - \gamma)^{2i}$. Also, $\|I + B_* B_*^T\| \ \leq\ 1 + \|B_*\|^2 \ \leq\ 1 + \beta^2$. We conclude that

$$\|\Sigma\| \ \leq\ (1 + \beta^2) \kappa^4 \sum_{i=0}^{\infty} (1 - \gamma)^{2i} \lesssim \beta^2 \kappa^4 \gamma^{-1}. \tag{101}$$

  Now, $x_t \sim \Sigma^{1/2} z$, where $z \sim N(0, I)$. Thus, with high probability, for all $t \in [T_0 + 1]$, $\|x_t\| \ \leq\ \widetilde{O}(\beta \kappa^2 \gamma^{-1/2}) \cdot \sqrt{d_x}$.

Since at each step we suffer regret at most $\widetilde{O}(\kappa^3 \beta \gamma^{-1/2} \sqrt{d_x})$ and the warm-up period is the interval $\{1, 2, \ldots, T_0\}$ and $T_0 = \widetilde{\Theta}(\kappa^{10} \beta^4 \gamma^{-5}) \cdot d_x d_u (d_x + d_u)^3$, we are done. $\square$

# G   Auxiliary Claims

**Claim 65.** *For all $K \in \mathcal{K}$, $\mathbb{E}_K[\|x_t^K\| + \|u_t^K\|] \leq \widetilde{O}(\kappa^3 \gamma^{-1/2} \sqrt{d_x})$.*

*Proof.* We have $x_t^K = (A_* + B_* K) x_{t-1}^K + w_{t-1} = \sum_{i=0}^{t-1} (A_* + B_* K)^i w_{t-i}$. Thus,

$$
(\mathbb{E}\|x_t\|)^2 \leq \mathbb{E}\|x_t\|^2 = \mathrm{tr}\left( \sum_{i=0}^{t-1} ((A_* + B_* K)^T)^i (A_* + B_* K)^i \right)
$$

$$
\leq d_x \sum_{i=0}^{t-1} \|A_* + B_* K\|^{2i}
$$

$$
\leq d_x \kappa^4 \sum_{i=0}^{\infty} (1 - \gamma)^{2i} \leq O(d_x \kappa^4 \gamma^{-1}). \tag{102}
$$

Thus, $\mathbb{E}\|x_t\| \leq \widetilde{O}(\kappa^2 \gamma^{-1/2} \sqrt{d_x})$.
Finally, we have $\mathbb{E}\|u_t\| = \mathbb{E}\|K x_t\| \leq \kappa \mathbb{E}\|x_t\| \leq \widetilde{O}(\kappa^3 \gamma^{-1/2} \sqrt{d_x})$. $\qquad\square$

**Claim 66.** *For all $M \in \mathcal{M}$, $R^{st}(M) \leq \widetilde{O}(\kappa^5 \theta^2 \gamma^{-2}) \cdot \sqrt{d_x}$.*

*Proof.* We use the quantity $z(M \mid A_*, B_*, \eta)$ that we define in 13.
$$
R^{st}(M) = \mathcal{C}(M \mid A_*, B_*) - \mathcal{C}(M_* \mid A_*, B_*)
$$
$$
\leq \mathbb{E}_\eta \|z(M \mid A_*, B_*, \eta) - z(M_* \mid A_*, B_*, \eta)\|
$$
$$
\leq \mathbb{E}_\eta \|z(M \mid A_*, B_*, \eta)\| + \mathbb{E}_\eta \|z(M_* \mid A_*, B_*, \eta)\|
$$
$$
\leq \widetilde{O}(\kappa^5 \beta^2 \gamma^{-2}) \cdot \sqrt{d_x}, \tag{103}
$$
where we used the definition of $\mathcal{C}$, that $c$ is 1-Lipschitz and Claim 55. $\qquad\square$

**Claim 67.** *With high probability, for all $t$,*

$$
\|\overline{z}_t - z_t\| \lesssim \kappa^5 \beta^2 \gamma^{-1} \cdot \sum_{i=1}^{2H+1} \|\widehat{w}_{t-i} - w_{t-i}\| + 1/T. \tag{104}
$$

*Proof.* First, for all $t$,
$$
\|\overline{u}_t - u_t\| = \left\| \sum_{i=1}^{H} M_t^{[i-1]}(w_{t-i} - \widehat{w}_{t-i}) \right\| \leq \sum_{i=1}^{H} \kappa^3 \beta (1-\gamma)^i \|w_{t-i} - \widehat{w}_{t-i}\|. \tag{105}
$$

Furthermore, $\|\overline{x}_t - x_t\| \leq \sum_{i=0}^{H} \|A_*^i\| \|B_*\| \|\overline{u}_{t-i-1} - u'_{t-i-1}\| + \|A_*^{H+1} x_{t-H-1}\|$. Claims 54 and 55 imply that with high probability we have $\|A_*^{H+1} x_{t-H-1}\| \leq \kappa^2 (1 - \gamma)^{H+1} \cdot \widetilde{O}(\kappa^5 \beta^2 \gamma^{-2}) \sqrt{d_x} \leq 1/T$. Using the bound 105, we get

$$
\|\overline{x}_t - x_t\| \leq \sum_{i=0}^{H} \kappa^2 \beta (1-\gamma)^i \sum_{j=1}^{H} \kappa^3 \beta (1-\gamma)^j \|w_{t-i-j-1} - \widehat{w}_{t-i-j-1}\| + 1/T
$$

$$
\leq \kappa^5 \beta^2 H \sum_{i=1}^{2H+1} \|w_{t-i} - \widehat{w}_{t-i}\| + 1/T. \tag{106}
$$

Finally,
$$
\|\overline{z}_t - z_t\| \leq \|\overline{x}_t - x_t\| + \|\overline{u}_t - u_t\|
$$

$$
\leq \cdot \kappa^5 \beta^2 H \sum_{i=1}^{2H+1} \|w_{t-i} - \widehat{w}_{t-i}\| + 1/T + \kappa^3 \beta \sum_{i=1}^{H} \|w_{t-i} - \widehat{w}_{t-i}\|
$$

$$
\leq \widetilde{O}(\kappa^5 \beta^2 \gamma^{-1}) \cdot \sum_{i=1}^{2H+1} \|w_{t-i} - \widehat{w}_{t-i}\| + 1/T. \tag{107}
$$

$\qquad\square$

**Claim 68.** *With high probability, for all t, we have* $\|\Delta_t\| \leq \gamma/(2\kappa^2)$.

*Proof.* The proof follows from Lemma 28, along with our choice of $T_0$. □

**Claim 69.** *For both Algorithms 1 and 4, we have that with high probability, for all t,*

$$c(x_t, u_t) - \mathcal{C}(M_t) \leq \widetilde{O}(\kappa^5 \beta^2 \gamma^{-2}) \cdot \sqrt{d_x}. \tag{108}$$

*Proof.*

$$
\begin{aligned}
c(x_t, u_t) - \mathcal{C}(M_t) &= c(z_t) - \mathbb{E}_\eta \left[ c \left( z(M \mid A_*, B_*, \eta) \right) \right] \\
&= \mathbb{E}_\eta \left[ c(z_t) - c \left( z(M \mid A_*, B_*, \eta) \right) \right] \\
&\leq \mathbb{E}_\eta \|z_t - z(M \mid A_*, B_*, \eta)\| \\
&\leq \|z_t\| + \mathbb{E}_\eta \|z(M \mid A_*, B_*, \eta)\|. 
\end{aligned} \tag{109}
$$

Lemma 60 and Claim 55 complete the proof. □

## G.1 Proof of Theorem 5

We will use $P_1, P_2, P_3$ to denote large polynomials in $T, d_x, \kappa, \beta, \gamma^{-1}$. Fix a $K \in \mathcal{K}$ and consider the corresponding dynamics $x_{t+1}^K = (A_* + B_* K) x_t^K + w_t$, $x_1^K = 0$. Also, $u_{t+1}^K = K x_{t+1}^K$. By unrolling the dynamics, we get $x_{t+1}^K = \sum_{i=0}^{t-1} (A_* + B_* K)^i w_{t-i}$. Thus,

$$u_{t+1}^K = K \sum_{i=0}^{t-1} (A_* + B_* K)^i w_{t-i}. \tag{110}$$

Let $M^{[i]} = K(A_* + B_* K)^i$, for $i = 0, 1, \ldots, H-1$. Also, consider the dynamics $u_{t+1}(M) = \sum_{i=0}^{H-1} M^{[i]} w_{t-i}$ and $x_{t+1}(M) = A_* x_t(M) + B_* u_t(M) + w_t$, $x_1(M) = 0$. We prove the following claim.

**Claim 70.** *For all t,* $\mathbb{E}_w \|u_{t+1}^K - u_{t+1}(M)\| \leq 1/P_1$, *where* $c_1$ *is a large constant and* $w$ *denotes the sequence* $(w_t)_t$.

*Proof.*

$$
\mathbb{E}_w \|u_{t+1}^K - u_{t+1}(M)\| \leq \mathbb{E}_w \|K \sum_{i=H}^{t-1} (A_* + B_* K)^i w_{t-i}\| \leq \sqrt{d_x} \sum_{i=H}^{t-1} \|K\| \cdot \|(A_* + B_* K)^i\|
$$

$$
\leq \kappa \sqrt{d_x} \sum_{i=H}^{t-1} \kappa^2 (1-\gamma)^i \leq \kappa^3 \sqrt{d_x} (1-\gamma)^H / \gamma \leq 1/P_1, \tag{111}
$$

because of the way we choose $H$. □

We have

$$x_{t+1}^K = \sum_{i=0}^{t-1} A_*^i w_{t-i} + \sum_{i=0}^{t-1} A_*^i B_* u_{t-i}^K \tag{112}$$

and

$$x_{t+1}(M) = \sum_{i=0}^{t-1} A_*^i w_{t-i} + \sum_{i=0}^{t-1} A_*^i B_* u_{t-i}(M) \tag{113}$$

By subtracting, we get

$$
\|x_{t+1}^K - x_{t+1}(M)\| \leq \sum_{i=0}^{t-1} \|A_*^i\| \cdot \|B_*\| \cdot \|u_{t-i}^K - u_{t-i}(M)\|
$$

$$
\leq 1/P_1 \cdot \sum_{i=0}^{t-1} \|A_*^i\| \cdot \|B_*\| \leq 1/P_2, \tag{114}
$$

Now, let $t \geq 2H + 2$, let $\eta(w) = (w_{t-1}, w_{t-1}, \ldots, w_{t-2H-1})$ and observe that $u(M \mid \eta(w)) = u_t(M)$ and $x(M \mid A_*, B_*, \eta(w)) = x_t(M) - A_*^{H+1} x_{t-1-H}(M)$. Thus,

$$\mathbb{E}_w \| x(M \mid A_*, B_*, \eta(w)) - x_t(M) \| \leq \| A_*^{H+1} \| \cdot \mathbb{E}_w \| x_{t-1-H}(M) \| \leq 1/P_3, \qquad (115)$$

since $\mathbb{E}_w \| x_{t-1-H}(M) \| \leq 1/P_2 + \mathbb{E}_w \| x_{t-1-H}^K \| \leq \widetilde{O}(\kappa^3 \gamma^{-1/2} \sqrt{d_x})$, by Claim 65. Overall, we have shown that for all $t \geq 2H + 2$,

$$\mathbb{E}_w \| u(M \mid \eta(w)) - u_t^K \| + \mathbb{E}_w \| x(M \mid A_*, B_*, \eta(w)) - x_t^K \| \leq 1/P_1 + 1/P_2 + 1/P_3. \qquad (116)$$

Using the above, we get

$$
\begin{aligned}
& |J(K) - \mathcal{C}(M \mid A_*, B_*)| \\
&= \left| \lim_{T \to \infty} \frac{1}{T} \cdot \mathbb{E}_w \sum_{t=1}^{T} c(x_t^K, u_t^K) - \mathbb{E}_w c\left(x\left(M \mid A_*, B_*, \eta(w)\right), u\left(M \mid \eta(w)\right)\right) \right| \\
&\leq \lim_{T \to \infty} \frac{1}{T} \cdot \sum_{t=1}^{T} \mathbb{E}_w \left( \| x_t^K - x(M \mid A_*, B_*, \eta(w)) \| + \| u_t^K - u(M \mid \eta(w)) \| \right) \\
&\leq 1/P_1 + 1/P_2 + 1/P_3 \leq 1/T. \qquad (117)
\end{aligned}
$$

### G.2 Proof of Theorem 8

We first define $C$-approximate barycentric spanners and cite a theorem that says that they can be computed in polynomial time given a linear optimization oracle. Given that theorem, it is easy to get the same result for the affine case, i.e., $C$-approximate affine barycentric spanners.

**Definition 71** ([5]). Let $K$ be a compact set in $\mathbb{R}^d$. A set $X = \{x_1, \ldots, x_d\} \subseteq K$ is a $C$-approximate barycentric spanner for $K$ if every $x \in K$ can be expressed as a linear combination of elements of $X$ using coefficients in $[-C, C]$.

**Theorem 72** (Proposition 2.5. in [5]). *Suppose $K \subseteq \mathbb{R}^d$ is compact and not contained in any proper linear subspace. Given an oracle for optimizing linear functions over $K$, for any $C > 1$ we can compute a $C$-approximate barycentric spanner for $K$ in polynomial time, using $O(d^2 \log_C(d))$ calls to the oracle.*

Let $x_0$ an arbitrary point in $K$ (we can get one with one call to the oracle). Let $K - x_0 := \{x - x_0 \mid x \in K\}$. Since $K$ is not contained in any proper affine subspace, $K - x_0$ is not contained in any proper linear subspace. Furthermore, the linear optimization oracle for $K$ is also a linear optimization oracle for $K - x_0$. Thus, Theorem 75 implies that for any $C > 1$ we can compute a $C$-approximate barycentric spanner for $K - x_0$ in polynomial time, using $O(d^2 \log_C(d))$ calls to the oracle, which finishes the proof.

### G.3 Proof of Theorem 5

We will use $P_1, P_2, P_3$ to denote large polynomials in $T, d_x, \kappa, \beta, \gamma^{-1}$. Fix a $K \in \mathcal{K}$ and consider the corresponding dynamics $x_{t+1}^K = (A_* + B_* K) x_t^K + w_t$, $x_1^K = 0$. Also, $u_{t+1}^K = K x_{t+1}^K$. By unrolling the dynamics, we get $x_{t+1}^K = \sum_{i=0}^{t-1} (A_* + B_* K)^i w_{t-i}$. Thus,

$$u_{t+1}^K = K \sum_{i=0}^{t-1} (A_* + B_* K)^i w_{t-i}. \qquad (118)$$

Let $M^{[i]} = K(A_* + B_* K)^i$, for $i = 0, 1, \ldots, H-1$. Also, consider the dynamics $u_{t+1}(M) = \sum_{i=0}^{H-1} M^{[i]} w_{t-i}$ and $x_{t+1}(M) = A_* x_t(M) + B_* u_t(M) + w_t$, $x_1(M) = 0$. We prove the following claim.

**Claim 73.** *For all $t$, $\mathbb{E}_w \| u_{t+1}^K - u_{t+1}(M) \| \leq 1/P_1$, where $c_1$ is a large constant and $w$ denotes the sequence $(w_t)_t$.*

*Proof.*

$$\mathbb{E}_w \| u_{t+1}^K - u_{t+1}(M) \| \; \leq \; \mathbb{E}_w \| K \sum_{i=H}^{t-1} (A_* + B_* K)^i w_{t-i} \| \; \leq \; \sqrt{d_x} \sum_{i=H}^{t-1} \| K \| \cdot \| (A_* + B_* K)^i \|$$

$$\leq \; \kappa \sqrt{d_x} \sum_{i=H}^{t-1} \kappa^2 (1-\gamma)^i \; \leq \; \kappa^3 \sqrt{d_x} (1-\gamma)^H / \gamma \; \leq \; 1/P_1, \qquad (119)$$

because of the way we choose $H$. $\qquad\square$

We have

$$x_{t+1}^K = \sum_{i=0}^{t-1} A_*^i w_{t-i} + \sum_{i=0}^{t-1} A_*^i B_* u_{t-i}^K \qquad (120)$$

and

$$x_{t+1}(M) = \sum_{i=0}^{t-1} A_*^i w_{t-i} + \sum_{i=0}^{t-1} A_*^i B_* u_{t-i}(M) \qquad (121)$$

By subtracting, we get

$$\mathbb{E}_w \| x_{t+1}^K - x_{t+1}(M) \| \; \leq \; \sum_{i=0}^{t-1} \| A_*^i \| \cdot \| B_* \| \cdot \mathbb{E}_w \| u_{t-i}^K - u_{t-i}(M) \|$$

$$\leq \; 1/P_1 \cdot \sum_{i=0}^{t-1} \| A_*^i \| \cdot \| B_* \| \; \leq \; 1/P_2, \qquad (122)$$

Now, we have

$$|J(K) - J(M)| = \left| \lim_{T \to \infty} \frac{1}{T} \cdot \mathbb{E}_w \sum_{t=1}^{T} \left( c(x_t^K, u_t^K) - c(x_t(M), u_t(M)) \right) \right| \qquad (123)$$

$$\leq \; \lim_{T \to \infty} \frac{1}{T} \cdot \mathbb{E}_w \left( \| x_t^K - x_t(M) \| + \| u_t^K - u_t(M) \| \right) \qquad (124)$$

$$\leq \; 1/P_1 + 1/P_2 \; \leq \; 1/T. \qquad (125)$$

### G.4 Proof of Theorem 6

We consider the dynamics $u_{t+1}(M) = \sum_{i=0}^{H-1} M^{[i]} w_{t-i}$ and $x_{t+1}(M) = A_* x_t(M) + B_* u_t(M) + w_t$, $x_1(M) = 0$. Now, let $t \geq 2H + 2$, $\eta(w) = (w_{t-1}, w_{t-1}, \ldots, w_{t-2H-1})$, and observe that $u(M \mid \eta(w)) = u_t(M)$ and $x(M \mid A_*, B_*, \eta(w)) = x_t(M) - A_*^{H+1} x_{t-1-H}(M)$. Thus,

$$\mathbb{E}_w \| x(M \mid A_*, B_*, \eta(w)) - x_t(M) \| \; \leq \; \| A_*^{H+1} \| \cdot \mathbb{E}_w \| x_{t-1-H}(M) \|. \qquad (126)$$

Now, in the proof of Theorem 5 (Appendix G.3), we showed that $\mathbb{E}_w \| x_{t-1-H}(M) - x_{t-1-H}^K \| \leq 1/P_2$, and in Appendix G (Claim 65), that $\mathbb{E}_w \| x_{t-1-H}^K \| \leq \widetilde{O}(\kappa^3 \gamma^{-1/2} \sqrt{d_x})$. So, we get that $\mathbb{E}_w \| x(M \mid A_*, B_*, \eta(w)) - x_t(M) \| \leq 1/P_3$. Using this bound, we get

$$|J(M) - \mathcal{C}(M \mid A_*, B_*)|$$

$$= \left| \lim_{T \to \infty} \frac{1}{T} \cdot \mathbb{E}_w \sum_{t=1}^{T} c(x_t(M), u_t(M)) - \mathbb{E}_w c \left( x \left( M \mid A_*, B_*, \eta(w) \right), u \left( M \mid \eta(w) \right) \right) \right|$$

$$\leq \; \lim_{T \to \infty} \frac{1}{T} \cdot \sum_{t=1}^{T} \mathbb{E}_w \left( \| x_t(M) - x(M \mid A_*, B_*, \eta(w)) \| + \| u_t(M) - u(M \mid \eta(w)) \| \right)$$

$$\leq \; 1/P_3 \; \leq \; 1/T. \qquad (127)$$

### G.5 Proof of Theorem 8

We first define $C$-approximate barycentric spanners and cite a theorem that says that they can be computed in polynomial time given a linear optimization oracle. Given that theorem, it is easy to get the same result for the affine case, i.e., $C$-approximate affine barycentric spanners.

**Definition 74** ([5]). Let $K$ be a compact set in $\mathbb{R}^d$. A set $X = \{x_1, \ldots, x_d\} \subseteq K$ is a $C$-approximate barycentric spanner for $K$ if every $x \in K$ can be expressed as a linear combination of elements of $X$ using coefficients in $[-C, C]$.

**Theorem 75** (Proposition 2.5. in [5]). *Suppose $K \subseteq \mathbb{R}^d$ is compact and not contained in any proper linear subspace. Given an oracle for optimizing linear functions over $K$, for any $C > 1$ we can compute a $C$-approximate barycentric spanner for $K$ in polynomial time, using $O(d^2 \log_C(d))$ calls to the oracle.*

Let $x_0$ an arbitrary point in $K$ (we can get one with one call to the oracle). Let $K - x_0 := \{x - x_0 \mid x \in K\}$. Since $K$ is not contained in any proper affine subspace, $K - x_0$ is not contained in any proper linear subspace. Furthermore, the linear optimization oracle for $K$ is also a linear optimization oracle for $K - x_0$. Thus, Theorem 75 implies that for any $C > 1$ we can compute a $C$-approximate barycentric spanner for $K - x_0$ in polynomial time, using $O(d^2 \log_C(d))$ calls to the oracle, which finishes the proof.

## H  Disturbance-based policies

We show that under the execution of disturbance-based policy $M$, we have

$$x_{t+1} = A_*^{H+1} x_{t-H} + \sum_{i=0}^{2H} \Psi_i(M \mid A_*, B_*) w_{t-i}, \tag{128}$$

where

$$\Psi_i(M \mid A_*, B_*) = A_*^i \, \mathbb{1}_{i \leq H} + \sum_{j=0}^{H} A_*^j B_* M^{[i-j-1]} \, \mathbb{1}_{i-j \in [1,H]}. \tag{129}$$

This formula was derived in [4] and we rederive it here for completeness.

$$
\begin{aligned}
x_{t+1} &= \sum_{i=0}^{H} A_*^i (w_{t-i} + B_* u_{t-i}) + A_*^{H+1} x_{t-H} \\
&= \sum_{i=0}^{H} A_*^i w_{t-i} + \sum_{i=0}^{H} A_*^i B_* \sum_{j=1}^{H} M^{[j-1]} w_{t-i-j} + A_*^{H+1} x_{t-H} \\
&= \sum_{i=0}^{H} A_*^i w_{t-i} + \sum_{\ell=0}^{2H} \sum_{i=0}^{H} A_*^i B_* M^{[\ell-i-1]} w_{t-\ell} \, \mathbb{1}_{\ell-i \in [1,H]} + A_*^{H+1} x_{t-H} \\
&= A_*^{H+1} x_{t-H} + \sum_{i=0}^{2H} \Psi_i(M \mid A_*, B_*) w_{t-i}.
\end{aligned}
$$

## Footnotes

[8]This means that there exists decomposition $\widehat{A} = Q\Lambda Q^{-1}$ with $\|\Lambda\| \leq 1 - \gamma/2$, and $\|Q\|, \|Q^{-1}\| \leq \kappa$.

[9]We set $w_t = 0$, for all $t \leq 0$.

[10]The symbol $\dagger$ denotes the pseudoinverse.

[11]Observe that $M_s$ is also random.

[12]In [3] they consider $\zeta_t$ independent but the analysis easily generalizes to the martingale condition that we use.

[13] This of $c$ as a large constant.

[14] In [3], there is a typo, because they write $s = 2 \cdot \frac{\sigma}{\gamma^2} \log n$. However, they fixed it in the journal version [2], where the formula for $s$ is the one we give here.

[15]We define $H_0 = \emptyset$.

[16]This is guaranteed to hold after the warm-up exploration (Lemma 63).

[17]This is guaranteed to hold after the warm-up exploration (Lemma 63).