[Reviews · NeurIPS 2020]

Review 1

Summary and Contributions: This paper introduces an algorithm for online control of stochastic LDS with a convex cost. Using barycentric spanners results in the literature, it suggests an adaptive exploration scheme to directly update controllers without going through the tedious control design procedures. It shows $\sqrt(T)$ regret result for both known or bandit-feedback convex cost functions. The novel contribution is the adaptive exploration algorithm.

Strengths: 1. The adaptive exploration algorithm is novel and can be of independent interest for general settings like partially observable or non-linear systems. The technical results are sound and handled cleanly. 2. Provides a time-optimal regret guarantee for online control of LDS with fixed convex cost and Gaussian disturbance. This result generalizes the result of $\sqrt(T)$ in strongly convex cost with semi-adversarial disturbance [20] and improves $T^2/3$ in convex cost with fully adversarial disturbances [20, 13]. 3. Instead of the ETC approach, the algorithm is fully adaptive and updates the system estimates and controller throughout the interaction which is more natural in online control settings.

Weaknesses: 1. The algorithm needs a known stabilizing controller or open-loop stability to achieve initial estimates of the system via naive exploration with Gaussian disturbances. This can be restrictive in some situations especially if the agent has no idea about the underlying system. Moreover, the assumption of Gaussian noise is fairly restrictive since in the recent online control of LDS results, this assumption is relaxed to more general family of disturbances and to adversarial disturbances. 2. The regret is surprisingly defined as comparison against the strongly stable linear controllers with fixed state feedback instead of best LDC controller with internal state which provides more fair comparison in terms of regret. I believe this weakens the result. Even though these are optimal in the classic LQR setting, as authors mention in the main text their optimality is uncertain in the setting of general convex cost functions. 3. The paper lacks numerical examples. This can be immensely valuable for this work by showcasing the polynomial-time computation of the algorithm in this general setting and validity of the results and limitations of the claims.

Correctness: The theoretical results are sound and the analysis is handled due diligence.

Clarity: The paper is clear and well-written. I believe the setting of bandit-feedback could be explained in more detail. Overall, critical intuitions and proof sketch is given in enough detail to guide the reader.

Relation to Prior Work: The work separates itself from prior work due to it's novel and more general setting of general convex results. The prior work is explained in detail.

Reproducibility: Yes

Additional Feedback: The paper provides an important tool to the literature of online control of LDS. Based on the mentioned strengths and weaknesses I am more inclined to accept since I believe the novel exploration algorithm can be useful in many settings for future research directions. However, I have some questions and suggestions to be included in the paper which would increase my score. 1. Can the choice of fixed state-feedback controller be justified in this setting of general convex loss functions? Do you have any intuitions on how the algorithm would compare against the best LDC controller with internal state which is considered in [20]? 2. Can the result be extended to general stochastic disturbances, e.g. sub-Gaussian disturbance or non-isotropic disturbances? 3. I believe the setting of [20] directly extends to partially observable LDS and results hold for both settings due to controller design with Disturbance Response Control. Would this exploration algorithm be adapted to partially observable LDS? 4. The arguments about running time are a bit unclear to me. Can we deduce that the oracle and all the required tools for execution of algorithm can be constructed in polynomial time? I believe this is very important feature of the suggested algorithm in terms of applicability. 5. Lastly, and maybe more importantly, it would be great to see some numerical results of the algorithm. I believe even for classical LQR setting with fixed p.d. Q and R, it would be good to see the performance of the algorithm and especially the effect of new exploration strategy. This will significantly increase the impact of the algorithm. ++++++++++After-rebuttal+++++++ Thanks for your response to the reviews. I still believe the lack of numerical examples (even in simple small LQR setting which should be easily doable) reduces the impact of the paper and brings out the question of the implementability of the algorithm. Even though the results should be considered fully technical as the authors mention, I believe this pure theoretical point of view without much implementability weakens the paper. Thus, I will keep my score at 6.


Review 2

Summary and Contributions: This paper gives order T^{1/2} regret for linear stochastic control with general cost convex costs in both the full information and bandit convex optimization settings.

Strengths: The results fill a subtle but important gap in learning for stochastic control. Namely, the regret was lowered in the setting of unknown dynamics with non-strongly convex costs. Many of the analytical methods seemed to be novel. I think this work is of definite relevance to the NeurIPS community. Edit: After reading the reviews and going back to the paper, I do agree that the methods are quite novel. For this reason I am bumping my score up.

Weaknesses: The biggest problem with the contribution is that it is not clear that the geometric exploration is really necessary, in light of the results on SBCO. Unless I am missing something major, the bandit case is quite disjoint from the full information case. Namely, while both cases have the same basic setup, the bandit case does not rely on the geometric exploration methodology. However, both get the same regret bounds, up to constant factors. So, aside from its methodological novelty, it is unclear if the geometric exploration algorithm gives any benefit beyond just reducing to a bandit algorithm. (Also, since the constant factors are not explicit, it is unclear which gives a better bound.) Edit: The reviewers have answered my question about the advantage of geometric exploration in their response. Namely, the dimension dependence is better. However, from reading the paper, I don't see how this can be deduced. The paper never gives a bound on the dimension dependence, other than saying that it is a large polynomial. This appears to be traced back to the analysis of the SBCO algorithm, from Theorem 45 and reference [3], which gives a bound in terms of an unspecified polynomial. Unless this is made more explicit in the paper / appendix, it definitely is not obvious how much better the geometric exploration method is.

Correctness: The ideas in the proof seem correct, but I admit that I could not follow the whole proof. It is very hard to read, as I will describe next.

Clarity: The main body of the paper is reasonably well written. The supplemental file is very hard to read. The proof proceeds in a highly nonlinear fashion. In many places, one claim will depend on two or more other claims, each of which is stated but then proved somewhere else in the text. There is no clear organization to where these proofs will arise. Stepping through the whole thing would take a massive amount of time. Also, the number of claims, theorems, and lemmas is mind-bogglingly high. The authors have gotten up to Theorem 71 in 34 pages of text. The text would not be so hard to read if the proofs were not so highly split. Most of the arguments are basic chains of inequalities using continuity and linear algebra. A proof that is a few pages long but has a clean flow would be much easier to read.

Relation to Prior Work: The authors do a good job of delineating their contributions compared to existing literature. However, as discussed above, they did not do a good job of delineating the relative merits of their own results when discussing the geometric exploration and the SBCO results. Edit: My issue here still remains. While the authors gave an explanation in their rebuttal, they did not give any indication that they intend to improve this for the paper.

Reproducibility: Yes

Additional Feedback: Edit: I do think that the reviewers should really attempt to streamline the appendix. Based on the other reviews, I am not the only person that had issues following the argument. However, I don't want to hold the paper up because of writing style.


Review 3

Summary and Contributions: - This is a purely theoretical paper that derives a regret bound for an online control problem. - The authors propose an geometric exploration scheme that employs the notion barycentric spanners, and use arguments from convex geometry to arrive at their regret bound - Their results apply to the case of a known cost function as well as to bandit feedback

Strengths: - The provided regret bound of T^1/2 improves upon the previous best known regret bound of T^2/3. - The theoretical contributions are quite involved, and include some ideas that could be more generally useful (e.g. the probabilistic coupling argument between the true system and the estimated one)

Weaknesses: The relevance of this paper is entirely unclear, for multiple reasons: 1. The author themselves state "This work does not present any foreseeable societal consequence.", raising the question why we should we care about this work in the first place. 2. They don't make any detectable effort towards arguing for why their work is relevant in the paper either, rendering it a purely theoretical exercise. 3. No empirical evaluation whatsoever is provided, there is no comparison (except for on an abstract level) with other methods. It is completely unclear what the practical value of the contribution even could be. Even a theoretical paper should at least try to argue for why it matters, this is not the case with this submission. The theoretical contributions may well be significant and valuable, however, in its current form this paper is not suitable for a publication at NeurIPS.

Correctness: It is hard for me to say, as I am not an expert and got lost in the details. However, I do not have reason to believe the theoretical claims are incorrect.

Clarity: * The lack of motivating examples or reference to practical problems makes paper inaccessible for anyone who is not an expert in the specific problem studied in the paper. * The presentation is very terse, the paper reads more like a list of bullet points rather than a well-formulated outline of research results. * Stating the results without ever giving even a high-level description of the algorithm is confusing.

Relation to Prior Work: Yes (though only by means of citing prior work without actually describing what that work does).

Reproducibility: Yes

Additional Feedback: - Typo: Lipscitz - The argument motivating general convex costs purely by means of state and control constraints is somewhat weak. There are other formulations for tackling this problem e.g. (MPC) that the authors do not discuss. - Some issues in the presentation: - "The above theorem" (theorem is below) - notation: K is used both for the controller and for denoting subsets of R^d. - at times hand-wavy arguments: "folklore approximation arguments suffice to show that ..." ******* Post author feedback *********** I don't have an issue with theoretical results at all, it is just the authors' complete lack of trying to make the paper somewhat accessible / point out the relevance to a reader at the margins / outside the field that is problematic. This submission is a great example for how to establish a cottage industry that isn't read outside a very small community. Also, for the record, I indicated that I was not very comfortable reviewing this paper to the AC after assignment, but the AC suggested I review as it's hard to find reviewers with relevant background. Finally, I appreciate the authors teaching me about the ancient greeks.


Review 4

Summary and Contributions: The paper presents the first optimal algorithm for online control of linear dynamical systems with i.i.d Gaussian noise, and unknown dynamics under general convex costs. There have been several recent works that looked at the same setting with adversarial noises and they have all employed a typical explore-then-commit style of approach which gives a regret guarantee of T^{-2/3}. It needs to be noted that they prove this for a broader problem setup than i.i.d gaussian noises that is considered in this paper. But there is no existing work that achieves optimal regret (sqrt{T}) in the i.i.d Gaussian noise setting with unknown dynamics and thus, this paper is novel in its contribution. The crux of the paper's contributions lies in the algorithm which performs adaptive exploration to identify the system dynamics instead of the typical explore-then-commit strategy (or certainty equivalence.) It is known from Stochastic Bandits literature that ETC approaches have suboptimal regret rate in T when compared to algorithms that do some sort of adaptive exploration such as Successive elimination/UCB that achieve optimal regret. In a similar fashion, this paper presents an algorithm that has an initial sysID phase to get reasonably close to the true dynamics (but not super close,) and then proceeds iteratively doing: first, it constructs a set of exploratory policies using the "barycentric spanners" of the convex set of policies (using a previously established disturbance-action policy class from Agarwal, Bullins, Hazan et. al.), it executes these exploratory policies to collect data that is used to refine the dynamics estimate and additionally, also use the executions to eliminate policies that have suboptimal performance (very similar to successive elimination condition.) Crucial to the algorithmic is the construction of barycentric spanners of the convex policy class which can be computed in polynomial time (assuming access to a linear optimization oracle -- which are typically efficient for convex classes using ellipsoid method) and the fact that these barycentric spanners can capture the geometry of the policy class (any policy in the policy class can be expressed as a linear combination of these spanners with bounded coefficients). When compared to ETC-style algorithms, the major advantage of the presented algorithm is the fact that exploration and elimination are interleaved leading to more efficient use of samples and better regret guarantees. This algorithm is applied in two settings: a full-information setting where the convex cost function is exposed to the learner, and a bandit setting where only the scalar cost is exposed. For the full-information setting, the algorithm follows elimination-style where at the end of each iteration a subset of current policy class is eliminated. For the bandit setting, the paper exploits robustness of existing approach of efficient stochastic convex optimization with bandit feedback (Agarwal, kakade, Rakhlin et. al.) In addition to showing the optimal regret of \sqrt{T} in both settings, the paper also argues that the algorithms can be implemented in polynomial time emphasizing that they can potentially be useful in practice as well. Similar to other optimization methods that exploit geometry (ellipsoid method etc.), the regret guarantee is not dimension-free and has strong dependence on the dimensionality of state space, action space and the size of the policy class (through H.) This dependence becomes stronger in the bandit setting which is expected. The core component of the proof relies on showing that the estimates of disturbance/noise used by the algorithm is close to the true disturbance with high probability. This is used to show that the cost of any policy under estimated dynamics is close to the cost of the same policy under true dynamics. Thus, any elimination done on the basis of a policy's performance under estimated dynamics can be shown to be valid under true dynamics as well with high probability.

Strengths: The strengths of the paper lie in its theoretical contributions: the idea of using geometry-based techniques such as barycentric spanners to achieve optimal regret, the new proof techniques used to show that the disturbance estimates lie close to true disturbances, and the regret analysis itself. Application of the idea to both full-information setting and the more realistic bandit setting is also a strength of the paper. In my opinion, the contribution of the paper is significant as it shows that the gap between upper and lower bounds can be closed in the challenging setting of online control with unknown dynamics and gaussian noise. I think extending it to the adversarial noise setting should be even more significant and is listed as a future work. I have some qualms regarding the practical applicability of the presented algorithms but keeping that aside, the contributions are significant in theory and new proof techniques The contributions are clearly novel as there exists no previous work that achieves optimal regret under this setting. They utilize the convex policy parameterization from previous work heavily and the idea of barycentric spanners from another previous work. The novelty of the paper is in identifying that we can achieve faster rates in convex optimization by exploiting geometry (like the ellipsoid method) and using the same ideas in the online control setting where the new parameterization lends itself into a convex problem. The work is relevant to NeurIPS community as it lies at the intersection of control, and online learning. This field has recently risen in its popularity among machine learning researchers who primarily work in online decision making.

Weaknesses: My major concern about the weakness of the contribution is similar to the weakness of geometry-based optimization methods: practical utility. As shown in the paper, the regret guarantee has a very strong dependence on the dimensionality of the problem. For even moderately sized state and action spaces, this guarantee can be potentially very weak. This is highly reflected in the lower bound on T in theorem 8, where the lower bound depends on the dimensionality of the problem by an order of 5! For a reasonably sized control problem with 5D state space, 3D action space this ends up being a constant of 8^5 which is huge. Thus, the given regret guarantee holds only after a very large number of executions are observed. There are also major concerns about the computational efficiency of the approach. The algorithm heavily relies on the idea that the barycentric spanners can be computed efficiently for convex policy classes. Although, they can be computed in polynomial time (as pointed out in the paper) practically this can still be very inefficient for large (convex) policy classes used in the paper. For an example, with a 5D state space, 3D action space and a horizon H=10, the number of parameters of any policy M is 10*3*5 = 150, which is really high for methods like ellipsoid methods. Thus, practically computing the barycentric spanners at each iteration of the algorithm can be computationally very expensive. I would've loved to see a small empirical experiment showing the computational efficiency (or lack of it) in the paper. Currently, the paper is mostly theoretical and presents a significant contribution in identifying that the optimal regret can be achieved. However, the computational efficiency of the algorithm is definitely in question even for moderately sized control problems and can be answered with one or two empirical experiments.

Correctness: As far as my knowledge goes (I am very familiar with online control but not so much about barycentric spanners), the claims and method seem to be correct. I have gone through the proofs on a high-level and they seem to be correct. No empirical methodology in the paper.

Clarity: Overall, the paper is well-written. I think the authors did a good job of presenting the basic ideas behind the algorithm and the proof. My major qualm with papers that are heavy in theory is that they seldom give a high-level intuition of the proofs. This paper, however, does a good job of giving a intuitive proof overview. To be a little nitpicky, I would've loved if the authors reduced the number of lemmas/theorems/corollaries in the main text and only identify the main theorems/lemmas and spend more space discussing the intuition behind the proof and theorem statements. The authors can improve the background section to include more explanation and intuition regarding the idea of barycentric spanners and what they lend to the presented algorithm. The idea that they can serve as good exploratory policies to learn the system dynamics needs more exposition as I think that is one of the paper's core contributions. More specifically, section 2.1.2 needs to be fleshed out to include this. As someone who was familiar with past works on online control, section 2.1.1 was good enough for me. But for someone who is not so familiar the background on disturbance-action policy class feels very rushed and cramped. This section can be greatly improved. There are several typos throughout the paper. Some of the major typos (which had me pondering for more than 5 mins) are: in line 177 (or algorithm 2) the loop index is s, but all the subscripts inside the loop are t (which can be confusing since Algorithm 1 had t and it invokes algorithm 2), and in line 159 (or definition 6) there seems to be a typo in the equation x = x_0 + \sum \lambda_i(x_i - x_0). I had to go read the original paper to get this.

Relation to Prior Work: Yes, the related/prior work does a reasonable job of explaining how this work differs from previous contributions.

Reproducibility: Yes

Additional Feedback: The original paper describes the barycentric spanners for a convex subset of R^d as a set of d elements, but this definition defines it as a set of d+1 elements. why is that? Can the lower bound on T in theorem 8 be improved? Or can you substantiate/explain how the regret guarantee is still useful for moderately sized control problems where it holds only after executing a very large number of executions. Can you discuss the practical computational efficiency of the presented algorithm? I understand that the computation is polynomial in T, but I am afraid that there can be a very huge dependence on dimensionality of the policy class and the state/action space. Can you analyze that dependence (and add to the paper?) I would love a comparison (either empirical or in terms of discussion) on the practical performance of the presented algorithm vs Hazan, Kakade, Singh paper on non-stochastic control. Can you give some intuition on this? I am guessing when the dimensionality of the control problem is very small, your algorithm might be more efficient but I am not sure how it would fare with moderately sized problems. *UPDATE after author feedback*: No update. I was happy with the paper before and the feedback did not affect my score.

[Author Response · NeurIPS 2020]

We thank the reviewers for their effort! Below we answer some of their questions.

General: we want to clarify that our results are purely theoretical, resolve an important question in exploration for control, and introduce techniques that we hope other researchers will build upon for practical applications.

**(R2) (necessity of geometric exploration)**
The value of the geometric exploration is that it achieves *much* better dependence in the dimension (not a constant factor!). Suppose $d=d\_x=d\_u$. The poly(d) in Theorem 2 (bandit case) is actually $d^{30}$ (!), while in Theorem 1 (geometric exploration) we have just $d^3$.
This is a major improvement over the bandit algorithm.
Since this was the *only* weakness presented in your review (other than the complaint on the organization of the Appendix), please consider updating your score.

**(R3) (relevance)**
1) We believe that our paper is highly relevant, since it tackles an important problem at the intersection of control theory and online learning, and follows up on numerous other works. As evidence, all the other reviewers said that this work has definite relevance.
2) If you cannot assess the value of a theoretical paper, please consider not reviewing it. Science is a long-term endeavor. It took millennia for number theory to evolve from a leisurely activity of the ancient Greeks to the basis of modern cryptography.

**(R1) (extensions of the result: best LDC, more general stochastic noise, partial observability)**
Yes, we can achieve all these extensions. We did not mention them in the paper, because we wanted to highlight our novel theoretical ideas. We will add one more section where we explain how all these extensions can be achieved.
1) the disturbance-based policies can express all stabilizing LDCs with internal state, so our regret bound actually holds wrt this richer policy class.
2) We can deal with any stochastic bounded disturbance (we need boundness for Lipschitz concentration to hold). The only place where the assumption of gaussian disturbances really helps is that given some policy M and matrices A, B, we can compute offline (to a very good approximation) the stationary cost C(M|A,B), because we know the disturbance distribution. However, even when we don't, we can still use the estimated disturbances (\hat{w}) as samples to approximate this expectation (i.e., the stationary cost).
3) The extension to partial observation is tedious but straightforward and uses the idea of "nature's y's", exactly as in [20].

**(R1) (poly time)**
Yes, we can prove polynomial time. We considered the implementation of the linear optimization oracle via the ellipsoid method to be folklore. We will add a formal proof at the Appendix.

**(R4) (dimension dependence running time)**
The overall dimension dependence in the runtime is $(d\_x*d\_u)^7$. We did not put effort into making this algorithm practical.

**(R1) (need for stabilizing controller)**
This is a standard assumption in online control. Without this assumption, the regret has an additive term which is exponential in the dimension (see the recent paper "Black box control for LDS" by Chen and Hazan).

**(R4) (Barycentric spanners, d vs d+1 elements)**
We use *affine* barycentric spanners and an affine basis has d+1 elements. We need this because the state is an affine function of the policies (not just linear).

Finally, we would like to thank reviewer 4 for their suggestions on the background section, which we will incorporate in the paper.

[Meta-Review · NeurIPS 2020]

The paper has received largely positive reviews (with the exception of one outlier), and the author response addressed the initial concerns adequately. Thus, the paper is suitable for publication at NeurIPS without significant changes, although the authors are strongly urged to implement the changes promised in the author response.